

# Dark matter from the centre of $SU(N)$

**Michele Frigerio[1], Nicolas Grimbaum-Yamamoto[2] and Thomas Hambye[2,3]**

**1** Laboratoire Charles Coulomb (L2C), University of Montpellier, CNRS, Montpellier, France
**2** Service de Physique Théorique, Université Libre de Bruxelles,
Boulevard du Triomphe, CP225, 1050 Brussels, Belgium
**3** CERN, Theoretical Physics Department, Geneva, Switzerland

## Abstract

A dark sector with non-abelian gauge symmetry provides a sound framework to justify stable dark matter (DM) candidates. We consider scalar fields charged under a $SU(N)$ gauge group, and show that the centre of $SU(N)$, the discrete subgroup $Z_N$ also known as $N$-ality, can ensure the stability of scalar DM particles. We analyse in some details two minimal DM models of this class, based on $SU(2)$ and $SU(3)$, respectively. These models have non-trivial patterns of spontaneous symmetry breaking, leading to distinctive phenomenological implications. For the $SU(2)$ model these include a specific interplay of two DM states, with the same interactions but different masses, and several complementary DM annihilation regimes, either within the dark sector or through the Higgs portal. The $SU(3)$ model predicts dark radiation made of a pair of dark photons with a unique gauge coupling, as well as regimes where DM semi-annihilations become dominant and testable.

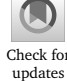

# 1 Introduction

In the Standard Model (SM) the four stable particles (photon, electron, lightest neutrino and proton) are all stable as a result of Lorentz and gauge invariance. The possibility that dark matter (DM) particles are stable for a similar reason has been extensively studied in the recent years. A local symmetry is preferable, in particular, because global symmetries can be broken by quantum gravity [1]. Gauge stabilisation of DM can be achieved either introducing a large (fermion or scalar) representation of the weak gauge group [2], or assuming the existence of a new gauge interaction in a dark sector. For fermion DM, the simplest hidden sector consists in nothing but a copy of QED, with a new $U(1)_D$ gauge symmetry [3,4] which may or may not be spontaneously broken. For vector DM, the simplest example is provided by a $SU(2)_D$ gauge symmetry spontaneously broken by a scalar doublet, so that the three gauge bosons are stable due to a remnant, accidental $SO(3)$ custodial symmetry [5,6]. The analogous construction for a $U(1)_D$ gauge symmetry can also lead to a stable gauge boson DM, but only assuming a somewhat ad hoc charge-conjugation symmetry, to forbid kinetic mixing with the hypercharge gauge boson [5,7,8]. For scalar DM, the simplest possibility is to assume a scalar DM field $\chi$ charged under a new $U(1)_D$ symmetry (with for definiteness unit charge), and that spontaneous symmetry breaking is induced by a second scalar field $\Phi$ with charge $n$, with $n$ for instance an integer larger than unity. In this case a $Z_n$ subgroup of $U(1)_D$ is unbroken and preserves the $\chi$ stability. Note it is straightforward (but not necessary) to make $\Phi$ and the gauge boson parametrically heavy, leaving $\chi$ as the lightest dark particle. This 'abelian' possibility has been considered e.g. in [9–12].

Here we focus on the alternative possibility of scalar DM stabilised by a non-abelian gauge symmetry of the dark sector, $G_D$. The non-abelian nature of the group $G_D$ comes with some theoretical advantages, such as asymptotic freedom and charge quantisation, as well as with a much richer phenomenology. Previous works [13,14] considered various Lie groups for the dark gauge symmetry, assuming a single scalar representation, either fundamental or with two indices. A recent paper [15] considered $SU(2)_D$ broken by a three-index scalar, i.e. a quadruplet. In these works various possible DM candidates were identified, due to the existence of some remnant gauge symmetry, or global accidental symmetry. When the SSB of $G_D$ preserves a non-abelian subgroup (or even when the entire $G_D$ is unbroken), the theory may confine in the infrared. In such case some of the resulting bound states may be stable DM candidates [6,13,14]. There is also the possibility to have stable scalar glueballs from the $G_D$ confinement, without assuming any dark scalar field [16,17]. On the other hand, a dark scalar allows for a Higgs portal interaction between the dark sector and the SM.

In this paper we call attention on a different possibility to realise scalar DM from a non-abelian gauge symmetry. We take advantage of the 'N-ality' symmetry of $SU(N)$ gauge theories,

which corresponds to the existence of a non-trivial group centre $Z_N$, formed by those transformations which commute with all $SU(N)$ elements. The idea is quite simple: when $SU(N)$ is spontaneously broken by a scalar representation $\Phi$ which is invariant with respect to $Z_N$, the centre subgroup remains unbroken. In this case, the lightest particles charged under the centre will be stable, and potential DM candidates. In particular, any scalar multiplet transforming non trivially with respect to $Z_N$ could provide in this way one or several DM candidates. Note that the $SU(N)$ centre may have other phenomenological applications in different contexts. For example the role of the centre of a $SU(N)$ gauge symmetry was explored in axion models, in connection with the quality of the Peccei-Quinn symmetry [18, 19]. A recent paper [20] also studied the center of discrete symmetry groups, and pointed out the possibility to use it to stabilise DM.

We will analyse two simple models of scalar DM protected by $N$-ality, based on an $SU(2)$ and $SU(3)$ gauge symmetry respectively, which have interesting theoretical properties and rich phenomenologies. On the theory side, the SSB pattern of these models preserves $U(1)$ gauge symmetries, as well as accidental global symmetries beside the center $Z_N$. In addition the scalar potential of the $SU(3)$ model turns out to have a vacuum manifold with an interesting structure. As for the dark sector phenomenology, its richness results mainly from the non-abelian structure which leads to multi-component DM with specific interplay between the various components. For the $SU(3)$ case the preserved triality symmetry $Z_3$, inherited from the non-abelian group, also allows for processes with an odd number of DM particles. To account for the observed DM relic density, we will mainly focus on the usual thermal freeze-out regime. Several freeze-out production regimes will be considered with specific phenomenology. Other production mechanisms will also be considered briefly.

The paper is structured as follows. In section 2 the properties of $N$-ality are reviewed and the scalar DM models based on $N$-ality are introduced. In section 3 the scalar potential for each model is introduced, and its minimisation is studied, while in section 4 we present the resulting set of dark-sector masses and couplings. While the discussion for the $SU(2)_D$ model is relatively compact, the analysis of the $SU(3)_D$ model is theoretically instructive, but technically involved: the reader mostly interested in DM phenomenology can skip the corresponding subsections. In section 5 we study the phenomenology of our models: the constraints on dark radiation, the computation of the DM relic density in various regimes, and finally the role of DM semi-annihilations.

## 2 SU(N) dark sector and $N$-ality

We will assume a single gauge group $G_D$ in the dark sector with coupling strength $g_D$. In the case $G_D = SU(N)$, a natural candidate for the symmetry stabilising DM is the so-called $N$-ality, that is, the $Z_N$ subgroup which constitutes the centre of $SU(N)$: each irreducible $SU(N)$ representation, with $n$ upper and $m$ lower indices, is assigned a $Z_N$ charge given by $(n-m)$ mod $N$. Defining $\omega = \exp(2i\pi/N)$, e.g. the fundamental representation transforms as $\omega$, the anti-fundamental as $\omega^{N-1}$, the adjoint as $\omega^0$, et cetera.

While $SU(N)$ invariance guarantees that the Lagrangian is $Z_N$ invariant, spontaneous symmetry breaking may or may not preserve such symmetry. For our program, the simplest possibility is to introduce two scalar fields, $\chi$ in the fundamental and $\Phi$ in the $N$-index symmetric representation, which is $Z_N$-invariant. Assuming that $\Phi$ acquires a VEV while $\chi$ does not, one obtains that $Z_N$ remains unbroken, and $\chi$ is automatically stable. With this choice of representations, the $Z_N$ symmetry is especially manifest in the coupling $\Phi^*_{i_1 \dots i_N} \chi^{i_1} \dots \chi^{i_N}$, which is non-vanishing as the $N$ indices of $\Phi$ are symmetrised and the scalars are commuting fields.

In the case of $SU(2)$, $\Phi \sim \mathbf{3}$ coincides with the adjoint, the potential includes a cubic coupling $\Phi_{ij}\chi^i\chi^j$, and the discrete DM symmetry is $Z_2$, the so-called $SU(2)$ 'duality'. In the case of $SU(3)$, $\Phi \sim \mathbf{10}$ is a 3-index symmetric tensor, the potential includes a quartic coupling $\Phi^*_{ijk}\chi^i\chi^j\chi^k$, and the DM symmetry is $Z_3$, the $SU(3)$ 'triality'.

We will not consider $SU(N)$ for $N > 3$. Note the coupling of $\Phi$ to $N$ copies of $\chi$ is non-renormalisable for $N \geq 4$. Moreover, the $SU(N)$ gauge theory with one scalar $\Phi$ in the $N$-index-symmetric representation loses asymptotic freedom already for $N \geq 5$: the relevant Dynkin index is $T(\Phi) = (2N)!/[2(N+1)!(N-1)!]$. We do not need to confront with these complications, since the DM phenomenology is very rich already for the cases $N = 2$ and $N = 3$.

It is worth remarking that analogous $SU(N)$ DM models could be built by changing the representation of $\chi$ and/or $\Phi$, provided the former carries a non-trivial $N$-ality while the latter does not. For example, one can (partially) break $SU(N)$ by the VEV of an adjoint scalar $\Phi^i_j$. For $N = 2$, the adjoint coincides with the two-index symmetric, and we will study this model in details below. For $N > 2$, the adjoint scenario is qualitatively different, and it allows an easier extrapolation to large $N$, in comparison to the case of $\Phi$ in the $N$-index symmetric. However, the role of $N$-ality is not manifest in the adjoint scenario, as there is no coupling of $\Phi$ to $N$ copies of $\chi$. The coupling $\chi^*_i \Phi^i_j \chi^j$ leads to a more traditional DM phenomenology, thus we will not further consider this possibility in this paper.

## 3 Spontaneous symmetry breaking in the dark sector

We are interested in a gauge group $G_D = SU(N)_D$ with a set of scalar fields in specific representations. The most general $SU(N)_D$-invariant, degree-four polynomial in the scalar fields defines the renormalisable scalar potential $V$ of the dark sector. Its minimisation determines the SSB pattern of the gauge symmetry, the mass spectrum of scalar and vector bosons, as well as the set of unbroken global symmetries.

### 3.1 The dark $SU(2)$ model

In the case of a dark $SU(2)_D$ with charge $g_D$, let us consider a real scalar triplet $\varphi^a$ where $a = 1, 2, 3$ is an index in the adjoint representation. The latter is equivalent to the two-index symmetric representation, as one can define a $2 \times 2$ symmetric matrix $\Phi$ with components

$$\Phi^{ij} \equiv \sqrt{2}\varphi^a(\tau^a)^i_k\epsilon^{kj}, \tag{1}$$

where $\tau^a \equiv \sigma^a/2$ are the $SU(2)$ generators and our convention for the Levi-Civita tensor is $\epsilon^{12} = -\epsilon_{12} = 1$. Notice that the reality condition reads $(\Phi^{ij})^* = \Phi_{ij} \equiv \epsilon_{ik}\Phi^{kl}\epsilon_{lj}$, and the normalisation is chosen such that $\Phi^{ij}\Phi_{ji} = \varphi^a\varphi^a$. The isospin eigenstates are given by $(\phi^+, \phi^0, \phi^-) = [(\varphi^1 - i\varphi^2)/\sqrt{2}, \varphi^3, (\varphi^1 + i\varphi^2)/\sqrt{2}] = (-\Phi^{11}, \sqrt{2}\Phi^{12}, \Phi^{22})$.

The most general renormalisable Lagrangian can be written as

$$\mathcal{L}(\Phi) = \frac{1}{2}D_\mu\Phi^{ij}D^\mu\Phi_{ij} - V(\Phi), \qquad V(\Phi) = -\frac{\mu^2}{2}\Phi^{ij}\Phi_{ji} + \frac{\lambda}{4}(\Phi^{ij}\Phi_{ji})^2. \tag{2}$$

The case of a $SU(2)$ gauge symmetry broken by a real scalar triplet is well known. For $\mu^2 > 0$ and $\lambda > 0$, the SSB pattern is $SU(2)_D \rightarrow U(1)_D$, driven by a VEV $\langle\Phi^{ij}\Phi_{ji}\rangle = \mu^2/\lambda \equiv v_D^2$. One can choose the VEV in the $\tau^3$ direction without loss of generality, and thus write $\varphi^3 = v_D + \rho$. The radial mode $\rho$ acquires mass $m_\rho^2 = 2\mu^2$, the gauge boson $W_D^\pm$ charged under $U(1)_D$ receives mass $m_{W_D}^2 = g_D^2\mu^2/\lambda$, while the neutral gauge boson $A_D$ remains massless.

We recall that such SSB pattern has a non-trivial second homotopy group, $\pi_2[SU(2)/U(1)] = Z$, corresponding to the existence of topologically stable monopoles [21, 22]. These may be produced in the early Universe, during the phase transition occurring at temperature $T_c \sim v_D$. In the case of a second order phase transition, the monopole relic density produced through the Kibble-Zurek mechanism can be estimated as $\Omega_{mon}h^2 \sim 1/g_D(T_c/150\text{TeV})^2$, where we followed [23] (see [24, 25] for some explicit models). This sets an upper bound on the scale $v_D$, in order not to overclose the Universe, and actually opens the possibility of monopole DM. On the other hand, the monopole relic density is negligible for e.g. $g_D \sim 1$ and $v_D \sim 10$ TeV, and it might be further diluted by monopole annihilations and/or later entropy injections; also, it would be extremely smaller in the case of first-order phase transition. We will neglect monopoles in the following.

Let us remark that the unbroken symmetry $U(1)_D$ is an accident due to the choice of a minimal scalar sector. It is conceivable that additional scalar multiplets may also acquire a VEV and complete the SSB of $SU(2)_D$, by making $A_D$ massive as well. Still, if only scalars with an even number of $SU(2)_D$ indices acquire a VEV, the $Z_2$ duality symmetry remains unbroken. For example one could consider a scalar sector formed by two real adjoints $\Phi_{1,2}^{ij}$, or by a four-index-symmetric representation $\Phi^{ijkl}$, a real quintuplet.

Let us now introduce a $Z_2$-odd scalar multiplet, that will provide a candidate for DM. The simplest possibility is a scalar $\chi^i$ in the fundamental representation of $SU(2)_D$, with potential

$$V(\chi) = \mu_\chi^2 \chi^i \tilde{\chi}_i + \lambda_\chi \left( \chi^i \tilde{\chi}_i \right)^2 , \tag{3}$$

where we defined $\tilde{\chi}_i \equiv (\chi^i)^*$. Here too we adopt the usual convention to lower and raise tensor indices, i.e. $\chi_i \equiv \epsilon_{ij}\chi^j$ and $\tilde{\chi}^j \equiv \epsilon^{jk}\tilde{\chi}_k$. The most general couplings between $\chi^i$ and a real adjoint scalar $\Phi^{ij}$ read

$$V(\Phi,\chi) = \frac{1}{2}(\kappa_1 \chi^i \Phi_{ij} \chi^j + h.c.) - \kappa_2 \chi^i \Phi_{ij} \tilde{\chi}^j + \frac{1}{2}\lambda_{\chi\Phi} \left( \Phi^{ij}\Phi_{ji} \right) \left( \chi^i \tilde{\chi}_i \right) . \tag{4}$$

Since there are no couplings linear in $\chi$, a vanishing VEV $\langle \chi \rangle = 0$ is automatically an extremum of the potential. In a large portion of the parameter space, $\langle \chi \rangle = 0$ is also a minimum.[1] As a consequence, the $Z_2$ symmetry $\chi \to -\chi$ is unbroken and protects its stability. In this region of parameters the VEV of $\Phi$ is determined by $V(\Phi)$ only, according to the discussion above.

A recent paper [26] considered a dark sector with the same gauge symmetry and scalar multiplets, but focusing on the region where also $\chi$ acquires a VEV, so the DM candidate and the symmetry responsible for its stability are different from ours. We note that vector (spin-one) DM is also possible in a dark sector with $SU(2)_D$ symmetry. Indeed, this is the case in minimal models with a single scalar multiplet: when $SU(2)_D$ is broken by one fundamental scalar $\chi^i$, the custodial triplet of gauge bosons $W_D^a$ is degenerate and stable [5]; when $SU(2)_D$ is instead broken by one adjoint scalar $\Phi^{ij}$, DM can be constituted by the complex gauge boson $W_D^\pm$ charged under the residual $U(1)_D$ symmetry [24]. Other models with vector DM, and also possibly additional scalar DM or glueball DM, from a dark $SU(3)$ gauge symmetry can be found in [27–29].

Let us further understand the symmetries of our model. In the limit where the couplings $\kappa_{1,2}$ are neglected, the potential is separately invariant under a global $SU(2)_{D\Phi}$ acting on $\Phi$ only, and a global $SU(2)_{D\chi}$ acting on $\chi$ only. Actually, similarly to what happens in the SM, $\chi$ enjoys a larger, custodial symmetry, $SU(2)_{D\chi} \times SU(2)_\chi$, acting on the $2 \times 2$ matrix $X \equiv (\tilde{\chi} \ \chi)$

---

[1]The (tree level) condition on the potential parameters to guarantee $\langle \chi \rangle = 0$ cannot be written in closed form, for the most general potential. In the limit of small cubic couplings, $\kappa_{1,2} \to 0$, one can show that such condition is given by $2\lambda\mu_\chi^2 > \lambda_{\chi\Phi}\mu^2$. Note we also require the potential to be bounded from below, which corresponds to the region $\lambda > 0$, $\lambda_\chi > 0$ and $\lambda_{\chi\Phi} > -2(\lambda\lambda_\chi)^{1/2}$.

as $X \to U_{D\chi} X (U_\chi)^\dagger$: indeed $\chi^i \tilde{\chi}_i = \text{tr}(X^\dagger X)/2$. To understand the impact of $\kappa_{1,2}$ on these global symmetries, it is worth considering a redefinition of the $\chi$ field, according to

$$\chi' \equiv \cos\theta\, \chi + \sin\theta\, \tilde{\chi}\,, \qquad \cos 2\theta = \frac{\kappa_2}{\sqrt{\kappa_1^2 + \kappa_2^2}}\,, \qquad \sin 2\theta = \frac{\kappa_1}{\sqrt{\kappa_1^2 + \kappa_2^2}}\,, \tag{5}$$

where we chose $\kappa_1$ real by appropriately rephasing the $\chi$ field, with no loss of generality. In components,

$$\chi' \equiv \begin{pmatrix} \chi_+ \\ \chi_- \end{pmatrix} = \begin{pmatrix} \cos\theta\, \chi^1 + \sin\theta\, \chi^{2*} \\ \cos\theta\, \chi^2 - \sin\theta\, \chi^{1*} \end{pmatrix}. \tag{6}$$

Note that this redefinition is compatible with $SU(2)_{D\chi}$, and $SU(2)_\chi$ is replaced by $SU(2)_{\chi'}$ defined analogously. Then, the potential (4) becomes (dropping from now on the prime on $\chi'$, for notational convenience)

$$V(\Phi, \chi) = -\kappa\, \chi^i \Phi_{ij} \tilde{\chi}^j + \frac{1}{2} \lambda_{\chi\Phi} \left( \Phi^{ij} \Phi_{ji} \right) \left( \chi^i \tilde{\chi}_i \right), \tag{7}$$

where $\kappa \equiv \sqrt{\kappa_1^2 + \kappa_2^2}$ is positive definite. Firstly, this shows that there is only one physical cubic coupling $\kappa$. Secondly, $\kappa$ breaks $SU(2)_{D\Phi} \times SU(2)_{D\chi}$ to a single $SU(2)_D$, corresponding to the gauged group we started from. Thirdly, $\kappa$ also breaks $SU(2)_\chi$ to the subgroup generated by $\tau_3$, which simply acts as $\chi \to e^{i\alpha_3/2} \chi$, $\tilde{\chi} \to e^{-i\alpha_3/2} \tilde{\chi}$. Thus, the model has an accidental $U(1)_\chi$ global symmetry, which contains the $Z_2$ duality as a subgroup. Note that SSB by $\langle \Phi \rangle \neq 0$ leaves $U(1)_\chi$ unbroken.

Finally the dark-sector scalars can communicate with the SM sector through the usual Higgs portal interactions,

$$V_{portal} = \left( \lambda_{\chi H} \chi^i \tilde{\chi}_i + \frac{1}{2} \lambda_{\Phi H} \Phi^{ij} \Phi_{ji} \right) H^\dagger H\,. \tag{8}$$

Note that the VEVs of $\Phi$ and $H$, $v_D^2 \equiv \langle \Phi^{ij} \Phi_{ji} \rangle$ and $v^2 \equiv 2\langle H^\dagger H \rangle$, are shifted by the coupling $\lambda_{\Phi H}$, but the VEV directions are not affected, therefore the pattern of SSB remains the same.

## 3.2 The dark $SU(3)$ model

In the case of a dark $SU(3)$ gauge symmetry, let us consider a scalar $\Phi^{ijk}$ in the three-index symmetric representation $\mathbf{10}$. Here $i, j, k = 1, 2, 3$ are indices in the fundamental representation of $SU(3)_D$. The conjugate $\Phi^*_{ijk}$ transforms in the $\overline{\mathbf{10}}$, with indices in the anti-fundamental.[2] Since the $\mathbf{10}$ is invariant with respect to the centre of $SU(3)_D$, its VEV always preserves the triality $Z_3$.

The possible $SU(3)$-invariant polynomials are obtained by contracting indices with the invariant tensors $\delta^i_j$, $\epsilon_{ijk}$ and $\epsilon^{ijk}$. The resulting, most general, renormalisable potential reads

$$\begin{aligned} V(\Phi) = &-\mu^2 \Phi^*_{ijk} \Phi^{ijk} + \lambda \left( \Phi^*_{ijk} \Phi^{ijk} \right)^2 + \delta\, \Phi^{i_1 j_1 k_1} \Phi^*_{i_1 j_1 k_2} \Phi^{i_2 j_2 k_2} \Phi^*_{i_2 j_2 k_1} \\ &+ \left( \eta\, \epsilon_{i_1 i_2 i_3} \epsilon_{j_1 j_2 j_3} \Phi^{i_1 j_1 k_1} \Phi^{i_2 j_2 k_2} \Phi^{i_3 j_3 k_3} \Phi^*_{k_1 k_2 k_3} + h.c. \right) \\ &+ \left( \sigma\, \epsilon_{i_1 j_2 k_3} \epsilon_{i_4 j_1 k_2} \epsilon_{i_3 j_4 k_1} \epsilon_{i_2 j_3 k_4} \Phi^{i_1 j_1 k_1} \Phi^{i_2 j_2 k_2} \Phi^{i_3 j_3 k_3} \Phi^{i_4 j_4 k_4} + h.c. \right). \end{aligned} \tag{9}$$

The terms on the first line are also invariant under an overall $U(1)$, $\Phi \to e^{i\alpha} \Phi$: if such $U(1)_D$ symmetry is also gauged, the quartic couplings $\eta$ and $\sigma$ are forbidden. Let us first focus on the limit $\eta, \sigma \to 0$, and note that in this case the potential can be rewritten as

$$V(\Phi) = -\mu^2 A^i_i + \lambda \left( A^i_i \right)^2 + \delta\, A^i_j A^j_i\,, \qquad A^i_j \equiv \Phi^{ikl} \Phi^*_{jkl}\,, \tag{10}$$

---

[2]This $SU(3)$ representation is reminiscent of the flavour ten-plet of baryons in QCD, formed by the ten symmetrised combinations of the $u$, $d$ and $s$ quarks.

where a sum over repeated indices is always understood. The traceless part of $A^i_j$ transforms in the adjoint of $SU(3)_D$, while $A^i_i$ is a singlet. Such potential for a **10** representation of $SU(3)$ was considered in [30], where a few remarkable properties were pointed out, in connection with the residual discrete symmetries after SSB. Here we present a more systematic analysis of the potential minimisation.

Note that the $SU(3)_D$ invariance allows to choose a basis where the matrix $A$ is diagonal, $A = diag(D_1, D_2, D_3)$, with $D_i \equiv \sum_{kl} |\Phi^{ikl}|^2 \geq 0$. This basis choice amounts to non-trivial relations among the ten independent components of $\Phi$, that is, $A^1_2 = A^1_3 = A^2_3 = 0$. With this trick, it becomes relatively straightforward to analyse SSB. Firstly, the potential $V$ is bounded from below if and only if the quadratic form $\lambda(\sum_i D_i)^2 + \delta \sum_i D_i^2$ is copositive-definite, that is, positive for all values $D_i > 0$. This occurs for

$$\lambda + \delta > 0 \quad \text{and} \quad 3\lambda + \delta > 0 \,. \tag{11}$$

The extrema of the potential must satisfy the equation $\partial V(\Phi)/\partial \Phi^*_{abc} = 0$, for all $a, b, c$. By carefully accounting for the multiplicity of the $\Phi$ components (e.g. $\Phi^{112}$, $\Phi^{121}$ and $\Phi^{211}$ are one and the same field), one can derive the extremality condition,

$$0 = \Phi^{abc}\left[-\mu^2 + 2\lambda(D_1 + D_2 + D_3) + \frac{2}{3}\delta(D_a + D_b + D_c)\right], \quad \text{for all } a, b, c \,, \tag{12}$$

where we adopted the basis with $A$ diagonal.

For $\mu^2 < 0$ there is only one extremum at the origin, $\Phi^{abc} = 0$, which is of course a global minimum with $SU(3)_D \times U(1)_D$ unbroken. In this case the $SU(3)_D$ confines at low energy, and possible DM candidates can be found among the lightest bound states. We do not investigate this possibility in this paper; various DM candidates in the confined phase of a dark gauge symmetry are discussed e.g. in [6], [13], [14].

For $\mu^2 > 0$ one can check there are various extrema, and their nature depends on the sign of $\delta$. For $\delta < 0$, the global minimum is obtained for $D_1 = D_2 = 0$ and $D_3 = \mu^2/[2(\lambda + \delta)]$, or equivalently for permutations of the indices $1, 2, 3$. This corresponds to all $\Phi$ components vanishing except for $|\Phi^{333}|^2 = D_3$. The SSB pattern is $SU(3) \times U(1) \rightarrow SU(2) \times U(1)$. There are 5 massive gauge bosons and 15 massive real scalars. Here it is the remnant $SU(2)$ group which confines at low energy, with some potentially stable bound states.

We will rather focus on the region $\mu^2 > 0$ and $\delta > 0$, where the unbroken gauge group turns out to be abelian. In this case the global minimum is obtained for

$$D_1 = D_2 = D_3 = D \equiv \frac{\mu^2}{2(3\lambda + \delta)} \,. \tag{13}$$

At the minimum the field $\Phi$ satisfies the matrix equation $\Phi^{ikl}\Phi^*_{jkl} = D\delta^i_j$. This amounts to 9 real conditions on 20 real degrees of freedom, so that the vacuum manifold is 11-dimensional. Quite surprisingly, this number is larger than the number of generators in $SU(3)_D \times U(1)_D$. It is possible to show[3] that the potential (10) has no accidental continuous symmetries, beside $SU(3)_D \times U(1)_D$, therefore there are at most 9 Nambu-Goldstone bosons (NGBs). Thus, remarkably, there are at least two flat directions in the vacuum manifold, which do *not* correspond to a gauge transformation.

Indeed, one can check that different points in the vacuum manifold correspond to different physical mass spectra, therefore they are not gauge-equivalent to each other. Let us present two relevant examples. (i) One solution is $|\Phi^{123}|^2 = D/2$ with all other $\Phi$ components vanishing. In this case the SSB pattern is $SU(3) \times U(1) \rightarrow U(1)_3 \times U(1)_8$, where the subscript stand for the two Cartan generators $\lambda_{3,8}$ in the $SU(3)$ algebra. Thus, there are 7 massive gauge bosons

---

[3]We thank Felix Brümmer for providing an explicit proof of that statement.

and 2 massless ones. Among the 20 real scalars contained in $\Phi$, 7 are the would-be NGBs eaten via the Higgs mechanism, 7 others are massive, and the remaining 6 are massless. (ii) Another solution is $|\Phi^{111}|^2 = |\Phi^{222}|^2 = |\Phi^{333}|^2 = D$ with all other components vanishing. In this case $SU(3) \times U(1)$ is fully broken, with 9 massive gauge bosons. Among the remaining 11 real scalars, 9 are massive and 2 are massless. Clearly, these two configurations are physically not-equivalent, even though they have the same (minimal) energy. One can also show that there are flat directions in the vacuum manifold connecting (i) to (ii), which correspond to scan over intermediate, not-equivalent field configurations.

To recapitulate, in the case $\mu^2 > 0$ and $\delta > 0$, the minimisation of Eq. (10) does not determine uniquely the physical minimum. There are various approaches to lift this degeneracy and identify the true minimum. One can go beyond the classical, renormalisable approximation: compute radiative corrections to be added to the tree-level potential, and/or add non-renormalisable operators. Alternatively, one can drop the gauged $U(1)_D$ symmetry, and restore the second and third lines of Eq. (9). Here we pursue the latter option, as our initial aim was to stabilise scalar DM with a purely non-abelian gauge symmetry.

Notice that, adding the $\eta$ and/or $\sigma$ quartic couplings, the potential is no longer a function of $A^i_j$ only, therefore the full characterisation of the extrema becomes a much harder task. Still, in the light of the previous discussion, one can make some ansatzes. If one considers the special point (ii), where $\Phi^{111,222,333} \neq 0$ only, one finds it is no longer an extremum, as soon as $\eta \neq 0$ or $\sigma \neq 0$. Let us focus, therefore, on the special point (i), with $\Phi^{123} \neq 0$ only. It is laborious but straightforward to check that, even when $\eta \neq 0$ and/or $\sigma \neq 0$, the extremality condition $\partial V(\Phi)/\partial \Phi^*_{abc} = 0$ is still satisfied for all $abc \neq 123$. For $abc = 123$, the extremality condition reads

$$0 = 6\Phi^{123}\left[-\mu^2 + 2|\Phi^{123}|^2(6\lambda + 2\delta + 8\sigma^* e^{-4i\alpha} + \eta e^{2i\alpha} + 3\eta^* e^{-2i\alpha})\right],\qquad (14)$$

where $\alpha$ is the VEV phase, $\Phi^{123} = |\Phi^{123}|e^{i\alpha}$, and the couplings $\sigma$ and $\eta$ are complex in general. Let us assume for simplicity both $\sigma$ and $\eta$ to be real, and $|\eta| > |2\sigma|$. In this region Eq. (14) has four solutions for the VEV of $\Phi^{123}$ (in addition to the trivial solution $\langle\Phi^{123}\rangle = 0$):

$$R: \quad \langle\Phi_{123}\rangle = \pm\sqrt{\frac{\mu^2}{4(3\lambda + \delta + 4\sigma + 2\eta)}}, \qquad I: \quad \langle\Phi_{123}\rangle = \pm i\sqrt{\frac{\mu^2}{4(3\lambda + \delta + 4\sigma - 2\eta)}}. \quad (15)$$

The sign ambiguity corresponds to a residual $Z_2$ symmetry of the potential, $\Phi \to -\Phi$, which implies that extrema come in degenerate pairs. In contrast, the solutions $R$ and $I$ are not degenerate, and they are related by the transformation $\eta \to -\eta$ and $\Phi \to i\Phi$.

Next, let us prove that a VEV for $\Phi^{123}$ not only provides an extremum, but also a minimum. The SSB pattern is $SU(3) \to U(1)_3 \times U(1)_8$, therefore there are 6 massive gauge bosons and, in the limit $\eta, \sigma \to 0$, 7 real scalars with positive mass and 7 massless ones. When one introduces $\eta \neq 0$, one can show that in the extremum $R$ the 7 massless scalars acquire a positive mass for $\eta < 0$ (when $\eta > 0$, these states have positive masses at the extremum $I$ instead). At the same time, the other 7 scalars retain a positive mass, as long as $|\eta|$ is not too large w.r.t. $\lambda$ and $\delta$. By continuity, all masses will remain positive also when $\sigma \neq 0$ is introduced, as long as the latter is sufficiently small. Therefore, either the extremum $R$ or $I$ is a minimum of the potential, in an appropriate interval of the parameters. For definiteness, we will study the phenomenology for $\eta < 0$, corresponding to a minimum in the extremum $R$, and we define $v_D^2 \equiv 12|\langle\Phi^{123}\rangle|^2$.[4] The case with $\eta > 0$ and the minimum in the extremum $I$ is physically equivalent.

We just proved that a scalar $\Phi$ in the **10** representation of $SU(3)_D$ may break the latter to $U(1)_3 \times U(1)_8$. As in the $SU(2)_D$ model, there are monopoles, since $\pi_2[SU(3)/U(1)^2] = Z \times Z$

---

[4]The normalisation $\Phi^{123} = (v_D + \rho + i\theta)/\sqrt{12}$ guarantees that the real scalars $\rho$ and $\theta$ are canonically normalised, given the kinetic term $\mathcal{L}_{kin} \equiv (D_\mu \Phi^*_{ijk})(D^\mu \Phi^{ijk})$.

is non-trivial, but we will neglect their relic density, according to the discussion of section 3.1. We will see in section 4.2 that the two massless gauge bosons form a doublet with respect to a residual discrete symmetry, and in section 5.1 that such 'dark photon' is compatible with cosmological constraints. Alternatively, one can conceive adding additional scalar multiplets to complete the $SU(3)_D$ SSB. In order to preserve the $Z_3$ triality, they should transform in representations $\hat{\Phi}^{i_1 \dots i_n}_{j_i \dots j_m}$ with $n - m = 0$ mod 3. The simplest examples are a second **10**, or an adjoint **8**. The DM phenomenology could be considerably different with and without massless dark photons, as it will be apparent by our analysis of the minimal model with a single **10** representation.

Let us now introduce a DM candidate protected by the triality $Z_3$. The simplest possibility is to consider a multiplet in the fundamental representation, $\chi^i \sim \mathbf{3}$. The latter has potential $V(\chi) = \mu_\chi^2 \chi^i \chi_i^* + \lambda_\chi (\chi^i \chi_i^*)^2$, where $\chi_i^* \sim \mathbf{\bar{3}}$ is simply the hermitian conjugate of $\chi^i$. The most general $\Phi - \chi$ couplings read

$$V(\Phi, \chi) = \left( \kappa\, \Phi^{ijk} \chi_i^* \chi_j^* \chi_k^* + h.c. \right) + \lambda_{\chi\Phi} \left( \Phi^{ijk} \Phi_{ijk}^* \right) \left( \chi^i \chi_i^* \right) + \lambda'_{\chi\Phi}\, \chi_i^* \Phi^{ijk} \Phi_{jkl}^* \chi^l\,. \tag{16}$$

In an appropriate, large region of the potential parameters, $\chi$ does not acquire VEV, and the VEV of $\Phi$ determined above is not perturbed by the interactions with $\chi$. There is no $\chi - \Phi$ mass mixing, the VEV of $\Phi$ preserves the triality $Z_3$, the $\chi$ stability is protected by its $Z_3$ charge, and $\chi$ is a good DM candidate.

All $\chi$ interactions preserve $\chi$-number, that is, they involve the same power of $\chi$ and $\chi^*$ fields, except for the quartic coupling $\kappa$, which involves three powers of $\chi$. Replacing $\Phi$ by its VEV, this corresponds to a DM cubic self-interaction $\chi^3$. In the limit where $\Phi^{ijk}$ masses are heavier than $m_\chi$, one can integrate out $\Phi$ and induce additional DM self-interactions, such as $\chi^3 \chi^{*3}$ or $\chi^4 \chi^*$. The DM has also gauge interactions with the $SU(3)_D$ gauge bosons, both the massive ones and the dark photons.

In section 4.2 we will derive explicitly the masses of all dark sector particles, as well as the relevant DM couplings. We will show that, after the $SU(3)_D$ SSB, a residual non-abelian symmetry is preserved. In particular, $(\chi^1\ \chi^2\ \chi^3)$ transform as a triplet under such symmetry. This implies that the three $\chi$ components carry one and the same mass, and share the same physical properties.

Finally, both $SU(3)_D$ scalars $\chi$ and $\Phi$ can communicate with the SM sector through the Higgs portal interactions,

$$V_{portal} = \lambda_{\chi H} \chi^i \chi_i^* H^\dagger H + \lambda_{\Phi H} \Phi^{ijk} \Phi_{ijk}^* H^\dagger H\,. \tag{17}$$

The VEVs of $\Phi$ and $H$, $v_D^2 \equiv 2\langle \Phi^{ijk} \Phi_{ijk}^* \rangle$ and $v^2 \equiv 2\langle H^\dagger H \rangle$, are shifted by the coupling $\lambda_{\Phi H}$, but the VEV directions and the SSB pattern remain unchanged.

# 4 Mass spectrum

## 4.1 The $SU(2)_D$ masses and couplings

Let us describe in detail the mass spectrum of the $SU(2)_D$ model defined by Eqs.(2)-(8). As already mentioned, the three gauge bosons split into the unbroken $U(1)_D$ gauge boson $A_D$, corresponding to the $\tau_3$ generator, and a complex gauge boson $W_D^\pm$ with unit dark charge, with masses

$$m_{A_D}^2 = 0\,, \qquad m_{W_D}^2 = g_D^2 v_D^2\,. \tag{18}$$

The real triplet scalar $\Phi^{ij}$ contains the two would-be NGBs, plus a radial mode $\rho$, neutral under $U(1)_D$. The latter mixes with the SM Higgs radial mode $h$, via the coupling $\lambda_{\Phi H}$ in

Eq. (8). Adopting the SM conventions $V(H) = -\mu_H^2 H^\dagger H + \lambda_H (H^\dagger H)^2$ with $H^T = [0 \, (v+h)/\sqrt{2}]$ and $v \simeq 246$ GeV, the $\rho - h$ mixing angle reads $\sin \theta_m \simeq \lambda_{\Phi H} v_D v / [2(\lambda_H v^2 - \lambda v_D^2)]$. The physical Higgs corresponds to the mass eigenstate $h_{phys} \simeq h - \sin \theta_m \rho$. Since the latter appears to be SM-like at the LHC, one needs roughly $|\sin \theta_m| \lesssim 0.2$, see e.g. [31]. Thus, in good approximation we can neglect order $\sin^2 \theta_m$ corrections to the mass eigenvalues, and simply obtain

$$m_\rho^2 \simeq 2\lambda v_D^2, \qquad m_h^2 \simeq 2\lambda_H v^2. \tag{19}$$

Coming to the scalar doublet of Eq.(6), its two complex components are distinguished by their $U(1)_D \times U(1)_\chi$ charges,

$$\chi_+ \sim \left( +\frac{1}{2}, +\frac{1}{2} \right), \qquad \chi_- \sim \left( -\frac{1}{2}, +\frac{1}{2} \right). \tag{20}$$

We recall from section 3.1 that these symmetries are a remnant of the custodial $SU(2)_D \times SU(2)_\chi$ symmetry of the complex doublet $\chi$: $SU(2)_D$ is broken spontaneously to $U(1)_D$ by the VEV of the triplet $\Phi$, while $SU(2)_\chi$ is broken explicitly to $U(1)_\chi$ by the cubic coupling $\kappa$ defined by Eq. (7). The $\chi$ components acquire masses

$$m_{\chi_\pm}^2 = \mu_\chi^2 + \frac{1}{2}\lambda_{\chi\Phi} v_D^2 + \frac{1}{2}\lambda_{\chi H} v^2 \pm \frac{1}{\sqrt{2}}\kappa v_D. \tag{21}$$

Note the accidental $U(1)_\chi$ prevents a mass mixing between $\chi_+$ and $\chi_-^*$. The coupling $\kappa$ is defined to be positive, it has mass dimension one, and it controls the mass splitting between $\chi_+$ and $\chi_-$,

$$\delta m_\chi^2 \equiv m_{\chi_+}^2 - m_{\chi_-}^2 = \sqrt{2}\kappa v_D. \tag{22}$$

In summary, the masses $m_{W_D}$, $m_\rho$, $m_{\chi_+}$, $m_{\chi_-}$, and $m_h$ depend on different couplings, and therefore they are independent, with the only constraint $m_{\chi_+} \geq m_{\chi_-}$.

Having derived the dark-sector mass spectrum, let us discuss the stability of the dark states. The massless dark photon $A_D$ is obviously stable, but not a DM candidate. The massive gauge boson $W_D^+$ carries unit charge with respect to $U(1)_D$, while it is neutral with respect to $U(1)_\chi$. As a consequence, its only possible decay channel is $W_D^+ \to \chi_+ \chi_-^* ...$, where dots stand for a set of particles which is neutral under $U(1)_D \times U(1)_\chi$. Therefore, $W_D$ is stable for $m_{W_D} \leq m_{\chi_+} + m_{\chi_-}$.

Coming to scalars, the radial mode $\rho$ is neutral with respect to all unbroken symmetries, and it has linear couplings both to other dark particles and to the Higgs boson. In particular, $\rho$ can always decay into SM particles through the Higgs portal.

Finally, let us discuss the stability of $\chi_\pm$, which are the states odd under the $SU(2)_D$ duality $Z_2$. They carry opposite $U(1)_D$ charge but the same $U(1)_\chi$ charge, according to Eq. (20). These symmetries highly restrict their couplings. In particular, one can check that the scalar potential only involves the combinations $\chi_\pm \chi_\pm^*$.[5] The combination $\chi_+ \chi_-^*$ only appears in the cubic coupling to $W_D^-$,

$$\mathcal{L} \supset \frac{i}{\sqrt{2}} g_D \left( \chi_-^* \partial^\mu \chi_+ - \chi_+ \partial^\mu \chi_-^* \right) W_{D\mu}^- + h.c. \tag{23}$$

Since all decays should preserve $U(1)_D \times U(1)_\chi$, the lightest state $\chi_-$ is always stable, and therefore a good DM candidate. Indeed, this was guaranteed from the start, since $\chi_-$ is the lightest odd particle under the $Z_2$ duality. In addition, it is easy to check that any $\chi_+$ decay must necessarily contain $W_D^+ \chi_-$ in the final state. However, such a transition is not kinematically

---

[5]The combination $\chi_+ \chi_-^*$ is not invariant with respect to $U(1)_D$, while $\chi_+ \chi_-$ is not invariant with respect to $U(1)_\chi$.

allowed if the $W_D$ mass is larger than the $\chi$ mass splitting, $m_{W_D} \geq m_{\chi_+} - m_{\chi_-}$. Even a virtual $W_D$ would have nowhere to go, therefore $\chi^+$ becomes an additional DM candidate in this case.

In summary, there are three possibilities for DM:

- DM content A: $\chi_-$ and $\chi_+$, for $m_{\chi_+} + m_{\chi_-} < m_{W_D}$;

- DM content B: $\chi_-$, $\chi_+$ and $W_D$, for $m_{\chi_+} - m_{\chi_-} \leq m_{W_D} \leq m_{\chi_+} + m_{\chi_-}$;

- DM content C: $\chi_-$ and $W_D$, for $m_{W_D} < m_{\chi_+} - m_{\chi_-}$.

This simple mechanism to have three different DM contents is a distinctive feature of the model. In this paper we will concentrate on the phenomenology of the scenario A, i.e. two scalar DM components, assuming the gauge boson is heavy enough to promptly decay.

Before concluding, we would like to stress that the $Z_2$ duality is sufficient by itself to guarantee the $\chi$ stability, and therefore a suitable DM candidate. As already explained, to preserve $Z_2$ is sufficient to sit in the 'half' of the potential parameter space where $\langle \chi \rangle = 0$. The additional unbroken symmetries $U(1)_D \times U(1)_\chi$ are due to the minimality of the model. Firstly, they would also be broken if $Z_2$ were broken by $\langle \chi \rangle \neq 0$. Secondly, they can be broken in less minimal models, which still preserve $Z_2$. As already mentioned in section 3, $U(1)_D$ can be broken spontaneously by the VEV of additional $Z_2$-even scalar multiplets, $\Phi'$. The $\Phi'$ couplings may also break explicitly $U(1)_\chi$. Alternatively, the accidental symmetry $U(1)_\chi$ might be broken by introducing higher-dimensional operators, induced by some UV physics.

In the case when the only unbroken symmetry is $Z_2$, all gauge bosons as well as $\Phi$ and $\Phi'$ components are massive and $Z_2$-even, therefore unstable. In contrast, the four real components of $\chi$ are $Z_2$-odd and have generically different masses: the lightest one, $\chi_D$, is a stable DM candidate. The limit where $\chi_D$ is much lighter than all other dark-sector particles correspond to the well-known SM-singlet scalar DM model [32–34]. Our gauged dark sector provides a rationale for the $Z_2$ parity, and predicts additional dark particles beside $\chi_D$.

## 4.2 The $SU(3)_D$ masses and couplings

Let us study the $SU(3)_D$ model around the minimum $R$ defined by Eq. (15). In order to study the masses and couplings of the physical multiplets, let us identify the relevant, unbroken symmetries after SSB. The unbroken continuous symmetries, $U(1)_3 \times U(1)_8$, act on a fundamental of $SU(3)$ as

$$Q_\alpha \equiv \exp(i\alpha\lambda_3) = diag(e^{i\alpha}, e^{-i\alpha}, 1), \qquad Q_\beta \equiv \exp(i\sqrt{3}\beta\lambda_8) = diag(e^{i\beta}, e^{i\beta}, e^{-2i\beta}). \quad (24)$$

In addition, there are unbroken discrete symmetries, which correspond to permutations of the three $SU(3)_D$ indices, generated by the matrices

$$P_3 \equiv \begin{pmatrix} 0 & 1 & 0 \\ 0 & 0 & 1 \\ 1 & 0 & 0 \end{pmatrix}, \qquad P_2 \equiv \begin{pmatrix} 0 & 1 & 0 \\ 1 & 0 & 0 \\ 0 & 0 & 1 \end{pmatrix}. \quad (25)$$

More precisely, $P_3$ generates the $Z_3$ group of even permutations, that are a subgroup of $SU(3)_D$ which leaves $\langle \Phi_{123} \rangle$ invariant. The $Z_2$ group generated by $P_2$ does not belong to $SU(3)_D$, nonetheless it is accidentally preserved by the most general $V(\Phi)$ in Eq. (9).[6] This $Z_2$ parity

---

[6]In general, the $SU(N)$ group contains the discrete subgroup $A_N$ of even permutations, as their determinant is $+1$, while the permutation group $S_N$ also contains odd permutations, with determinant $-1$. Since the covariant tensor $\epsilon_{i_1 \dots i_N}$ changes sign under a odd permutation, the $SU(N)$ invariants which involve an odd number of $\epsilon$'s are not invariant under $S_N$. In the present case, the most general potential (9) contains only couplings which involve an even number of $\epsilon$'s, hence it is invariant under the whole $S_3$.

also leaves $\langle \Phi_{123} \rangle$ invariant. Combining $P_3$ and $P_2$ one obtains the non-abelian group $S_3$ of all permutations of three indices.

Interestingly, the combination of $Q_{\alpha,\beta}$ and $P_{3,2}$ generates a hybrid non-abelian group $K$, partly continuous and partly discrete. The physical states of the model transform in specific representations of $K$, which can be simply characterised by the action of $Q_{\alpha,\beta}$ and $P_{3,2}$ on each given state.

The $SU(3)_D$ gauge bosons transform as $G_A^\mu \lambda_A \to U G_A^\mu \lambda_A U^\dagger$, for $U = Q_{\alpha,\beta}, P_{3,2}$. One can check that they decompose into a real $K$-doublet and a complex $K$-triplet, defined by

$$
A_D^\mu = \begin{pmatrix} G_3^\mu \\ G_8^\mu \end{pmatrix}, \qquad W_D^\mu = \frac{1}{\sqrt{2}} \begin{pmatrix} G_6^\mu + i G_7^\mu \\ G_4^\mu - i G_5^\mu \\ G_1^\mu + i G_2^\mu \end{pmatrix}. \tag{26}
$$

In particular, $A_D$ is neutral under $U(1)_3 \times U(1)_8$ and transforms as a doublet with respect to $S_3$, while $W_D$ components carry $U(1)_3 \times U(1)_8$ charges as well. The gauge boson masses are given by

$$
m_{A_D} = 0, \qquad m_{W_D}^2 = g_D^2 v_D^2. \tag{27}
$$

Interestingly, $A_D^\mu$ is a massless vector containing four physical degrees of freedom, i.e. a "four-component photon"!

Coming to the $SU(3)_D$ scalar ten-plet $\Phi$, its components transform under the group $K$ according to $\Phi^{ijk} \to U_a^i U_b^j U_c^k \Phi^{abc}$. One can check that the ten components organise themselves into $K$-multiplets according to

$$
\Phi^{123} = \frac{v_D + \rho + i\theta}{\sqrt{12}}, \qquad \tau = \begin{pmatrix} \Phi^{111} \\ \Phi^{222} \\ \Phi^{333} \end{pmatrix},
$$
$$
\varphi = \sqrt{\frac{3}{2}} \begin{pmatrix} \Phi^{133} + i\Phi_{122}^* \\ \Phi^{112} + i\Phi_{233}^* \\ \Phi^{223} + i\Phi_{113}^* \end{pmatrix}, \qquad \pi = \sqrt{\frac{3}{2}} \begin{pmatrix} i\Phi^{133} + \Phi_{122}^* \\ i\Phi^{112} + \Phi_{233}^* \\ i\Phi^{223} + \Phi_{113}^* \end{pmatrix}. \tag{28}
$$

The fields $\rho$ and $\theta$ are real $K$-singlets, while $\tau$, $\varphi_d$ and $\pi$ are complex $K$-triplets, all normalised to receive canonical kinetic terms from $\mathcal{L}_{kin}(\Phi) \equiv (D_\mu \Phi_{ijk}^*)(D^\mu \Phi^{ijk})$. More precisely, $K$ has different triplet representations depending on the $U(1)_3 \times U(1)_8$ charges: the $\tau$-triplet components transform with phases $(3\alpha + 3\beta, -3\alpha + 3\beta, -6\beta)$; in contrast, the three components of $W_D$, $\varphi$ and $\pi$ transform with phases $(\alpha - 3\beta, \alpha + 3\beta, -2\alpha)$. As a matter of fact, $\pi$ is the would-be NGB multiplet, eaten by the $W_D$ multiplet.

The non-Goldstone components of $\Phi$ acquire a mass according to

$$
\begin{aligned}
m_\rho^2 &= \frac{2}{3}(3\lambda + \delta + 4\sigma + 2\eta)v_D^2, & m_\theta^2 &= -\frac{2}{3}(8\sigma + \eta)v_D^2, \\
m_\tau^2 &= -\frac{1}{3}(4\sigma + 5\eta)v_D^2, & m_\varphi^2 &= \frac{2}{9}(2\delta - 12\sigma - 3\eta)v_D^2,
\end{aligned} \tag{29}
$$

which are all positive for appropriate choices of the quartic couplings, see the discussion in section 3.2. Here we adopted again the minimum $R$ defined by Eq. (15), and we neglected the contribution to these masses from the Higgs VEV, coming from the portal $\lambda_{\Phi H}$ defined by Eq. (17). In close analogy with our discussion for the $SU(2)_D$ model, this portal induces a $\rho - h$ mixing, which should be relatively small to respect Higgs constraints, setting an upper bound on $\lambda_{\Phi H}$. As the masses in Eq. (29) are already independent from one another, the Higgs-VEV corrections have no qualitatively relevant effects.

Finally, the $SU(3)_D$ scalar triplet $\chi$ transforms under the $K$ generators of Eqs.(24)-(25) as $\chi^i \to U^i_j \chi^j$, which corresponds to yet another $K$-triplet representation. Its mass is given by

$$m_\chi^2 = \mu_\chi^2 + \frac{1}{6}(3\lambda_{\chi\Phi} + \lambda'_{\chi\Phi})v_D^2 + \frac{1}{2}\lambda_{\chi H}v^2 \,. \tag{30}$$

Unlike for the $SU(2)_D$ model, the three components of $\chi$ are not split in mass, thanks to the unbroken $K$ symmetry. It is important to remark that this mass degeneracy is not a tree-level accident, that could be split e.g. by gauge boson loops. Quantum corrections do shift the various masses as usual, but all components of a $K$-triplet are equally shifted, because $K$ is an exact symmetry of the whole Lagrangian. Beside carrying $K$-charges, $\chi$ is also charged under the triality $Z_3$, which guarantees its stability and makes it a DM candidate.

By inspecting Eqs.(27), (29) and (30), one notices that $W_D$, the various $\Phi$ components and $\chi$ have independent masses, whose ordering is not fixed by the model. We will focus on the regime where $\chi$ is the lightest, and all other states rapidly decay into $\chi$ particles. Indeed, the $SU(3)_D$ gauge coupling $g_D$ allows for $W_D \to \chi\chi^*$, and the scalar potential couplings in Eq. (16) allow for, respectively,

$$\begin{aligned} \kappa : \;\; & \tau \to \chi^*\chi^*\chi^* \,, && \rho,\theta,\varphi \to \chi\chi\chi, \chi^*\chi^*\chi^* \,, \\ \lambda_{\chi\Phi} : \;\; & \rho \to \chi\chi^* \,, && \lambda'_{\chi\Phi} : \;\; \rho,\varphi \to \chi\chi^* \,. \end{aligned} \tag{31}$$

On the other hand, if $\chi$ were not sufficiently light for these decays to happen, then some of the states $W_D$, $\tau$ and $\varphi$ might be stable, as they carry $K$ charges (in contrast with $\rho$ and $\theta$ which are singlets). Then, they could provide additional DM components. We do not elaborate further on this possibility, and assume rapid decays of $W_D$ and of all $\Phi$ components.

In the limit where $W_D$ and $\Phi$ are heavy, the dark sector reduces to the triplet DM candidate $\chi$ and the doublet dark photon $A_D$. There are three type of interactions relevant for DM phenomenology. The $\chi - A_D$ gauge interactions read

$$\mathcal{L}_{kin} = \sum_{i=1}^{3} \left| \partial^\mu \chi^i - \frac{i}{2}g_D \left( G_3^\mu \lambda_3 + G_8^\mu \lambda_8 \right)^i_j \chi^j \right|^2 \,. \tag{32}$$

The effect of dark radiation on DM phenomenology is similar to the one in the $SU(2)_D$ model, up to the doubling of the dark photon components. The $\chi$ self-interactions read

$$\mathcal{L}_{self} = \sqrt{3}\kappa v_D \left( \chi^1\chi^2\chi^3 + \chi_1^*\chi_2^*\chi_3^* \right) + \lambda_\chi (\chi^i\chi_i^*)^2 \,. \tag{33}$$

The cubic self-interaction is the most specific feature of this scenario, which follows from $Z_3$ triality, and we will explore its phenomenological consequences below. Finally, the $\chi - h$ portal interactions read

$$\mathcal{L}_{portal} = -\lambda_{\chi H}(\chi^i\chi_i^*)\left(vh + \frac{1}{2}h^2\right) \,. \tag{34}$$

Here the Higgs portal phenomenology is completely standard, in contrast with the $SU(2)_D$ model which features two DM candidates with different masses.

## 5 Dark matter phenomenology

The phenomenological implications of the models are diverse. Here we will discuss the most straightforward case, where it is assumed that both visible and hidden sectors thermalise at early times. Only in section 5.2.4 we will depart from this assumption.

## 5.1 Dark radiation

The extra dark photons associated to the remnant $U(1)$ gauge symmetries imply extra radiation in the Universe. This is constrained from both BBN and CMB data. This extra radiation is in general parameterised in terms of the effective number of extra neutrinos. The dark photons induce

$$\Delta N_{eff} = \frac{8}{7}\left(\frac{T^0}{T_\nu^0}\right)^4 \frac{\rho_{\gamma_D}^0}{\rho_\gamma^0},$$ (35)

where $T \equiv T_\gamma$ and $T_\nu$ refer to the temperature of photons and neutrinos, $\rho_{\gamma,\gamma_D}$ refers to the energy density of photons and dark photons, and the index '0' stands for 'today'. The observational constraints on $N_{eff}$ are [35] $N_{eff}^{CMB} = 2.764_{-0.282}^{+0.308}$ and $N_{eff}^{BBN} = 2.878_{-0.226}^{+0.232}$. Subtracting the SM contribution, $N_{eff}^{SM} = 3.044$, one obtain at the 2 sigma level

$$\Delta N_{eff}^{CMB} < 0.336,$$ (36)

$$\Delta N_{eff}^{BBN} < 0.298,$$ (37)

$$\Delta N_{eff}^{CMB+BBN} < 0.135,$$ (38)

where the most stringent upper limit results from a combination of both types of constraint [35].

After the visible and the dark sectors decouple from each other, each sector can be reheated by particles decoupling,

$$\frac{T_D^0}{T^0} = \frac{T_D}{T}\left[\frac{g_D^{\star s}(T_D)}{g_D^{\star s}(T_D^0)}\right]^{1/3}\left[\frac{g_{SM}^{\star s}(T)}{g_{SM}^{\star s}(T^0)}\right]^{-1/3},$$ (39)

where $T_D$ refers to the temperature of the dark sector and $g^{\star s}$ are the effective number of relativistic degrees of freedom entering into the entropy density. We will take $g_{SM}^{\star s}(T)$ from [36]. Therefore, using also $\rho_{\gamma_D}^0/\rho_\gamma^0 = (g_{\gamma_D}^\star/g_\gamma^\star)(T_D^0/T^0)^4$, one obtains

$$\Delta N_{eff} = \frac{8}{7}\left(\frac{11}{4}\right)^{4/3}\frac{g_{\gamma_D}^\star}{g_\gamma^\star}\left[\frac{g_D^{\star s}(T_D^{dec})}{g_D^{\star s}(T_D^0)}\right]^{4/3}\left[\frac{g_{SM}^{\star s}(T^{dec})}{g_{SM}^{\star s}(T^0)}\right]^{-4/3}.$$ (40)

Here $T^{dec}$ refers to the temperature of the thermal bath when both sectors decouple from each other, with $T_D^{dec} = T^{dec}$, while $g_\gamma^\star = 2$ and $g_{\gamma_D}^\star$ are the number of degrees of freedom in photons and dark photons, respectively.

In the models we consider one can assume that the dark sector temperature is not reheated after both sectors decouple from each other,[7] so that the first square bracket in Eq. (40) is unity. For one dark photon, i.e. $g_D^\star = 2$, and using also $g_\gamma^\star(T^0) = 2$ and $g_{SM}^{\star s}(T^0) = 43/11$, one obtains

$$\Delta N_{eff}^{\gamma_D} = \frac{8}{7}\left(\frac{11}{4}\right)^{4/3}\left[\frac{43}{11 g_{SM}^{\star s}(T^{dec})}\right]^{4/3} = 0.0535\left[\frac{106.75}{g_{SM}^{\star s}(T^{dec})}\right]^{4/3}.$$ (41)

The reference value $g_{SM}^{\star s}(T^{dec}) = 106.75$ holds when the two sectors decouple before the SM particles have decoupled, i.e. for $T^{dec} \gtrsim 200$ GeV. Thus, at the 2 sigma level, the combined CMB and BBN constraint of Eq. (38) allows up to two dark photons, as long as $g_{SM}^{\star s}(T^{dec}) \sim 10^2$, that is to say, as long as the decoupling occurs before the QCD phase transition, i.e. for $T^{dec} \gtrsim 1$ GeV.

---

[7]This holds if both sectors decouple when the DM particle undergoes its freeze-out, which will be in general the case if the DM particle is the lightest massive particle in the dark sector. The dark photons are also expected to decouple at DM decoupling, as they communicate with the SM through the DM only.

In more details, requiring $\Delta N_{eff}$ not to exceed 0.135, as in Eq. (38), in the presence of one (two) dark photon(s) one needs[8]

$$g_{SM}^{\star s}(T^{dec}) \gtrsim 53 \rightarrow T^{dec} \gtrsim 300 \text{ MeV} \quad \text{(one dark photon)}, \tag{42}$$

$$g_{SM}^{\star s}(T^{dec}) \gtrsim 89 \rightarrow T^{dec} \gtrsim 30 \text{ GeV} \quad \text{(two dark photons)}. \tag{43}$$

The two cases correspond, indeed, to our minimal $SU(2)_D$ and $SU(3)_D$ models, respectively. Note that the future CMB-S4 ground based experiment could reach a precision of $\Delta N_{eff} = 0.03$ at the 1 sigma level [37], which would basically allow to determine the number of dark photons.

Note finally that both in the $SU(2)_D$ and $SU(3)_D$ models the dark photons do not kinematically mix with the SM hypercharge gauge boson, as a result of their non-abelian origin.

## 5.2 $SU(2)_D$ relic density and constraints

In the following we will determine the relic density for the case where the DM is made of $\chi_-$ and $\chi_+$. This corresponds to the DM particle content A, as defined in section 4.1, which corresponds to the region $m_{\chi_+} + m_{\chi_-} < m_{W_D}$, while the dark scalar $\rho$ could be lighter than $\chi_\pm$, and the dark photon $\gamma_D$ is massless.

The four types of $\chi$ interactions which contribute to DM annihilation are the gauge coupling $\alpha_D \equiv g_D^2/(4\pi)$ to $\gamma_D$, the cubic coupling $\kappa$ to $\rho$, and the $\lambda_{\chi\Phi}$ and $\lambda_{\chi H}$ quartic couplings to $\rho$ and $h$, respectively. Each of these interactions induces a DM annihilation process by itself. The annihilations proceed into a pair of dark photons (first case), a pair of $\rho$ particles if kinematically allowed (second and third case), and a pair of SM particles (fourth case). The first three cases are purely hidden sector processes, whereas the last one relies on the DM Higgs portal interaction with the SM. Various annihilation processes can also result from a combination of these interactions, as well as from the fact that the $\rho$ and $h$ scalars undergo a mass mixing induced by the $\lambda_{\Phi H}$ coupling. Fig. 1 shows the full list of $\chi_\pm$ annihilation diagrams. This also includes the annihilation of $\chi_+$ pairs into $\chi_-$ pairs.

In the following we will limit ourselves to three regimes where the DM annihilation is dominated by a single type of $\chi_\pm$ interaction, namely by $\alpha_D$, $\lambda_{\chi\Phi}$ or $\lambda_{\chi H}$. For simplicity, for these three regimes we will assume a small value of $\lambda_{\Phi H}$, so that the effect of $\rho$-$h$ mixing is negligible. These limiting cases will allow to illustrate the range of possibilities that the $SU(2)_D$ model offers, even if intermediate regions of parameters exist, where several annihilation channels compete with each other.

### 5.2.1 DM annihilation into dark photons

If the dark gauge coupling $\alpha_D$ dominates over other couplings, the $\chi_\pm$ annihilation proceeds dominantly into a pair of dark photons, see the first two diagrams of Fig. 1. The annihilation cross section for each DM component $\chi_\pm$ is given by (keeping the dominant s-wave contribution, see e.g. [38])

$$\langle \sigma v \rangle_\pm \equiv \langle \sigma_{\chi_\pm \chi_\pm^* \to \gamma_D \gamma_D} v \rangle = \frac{\pi \alpha_D^2}{4 m_{\chi_\pm}^2}, \tag{44}$$

where we took into account the factor coming from the DM charges, $Q_D^4 = 1/16$.

Imposing that the thermal freeze-out of these processes accounts for the observed DM relic density, requires in first approximation $(\langle \sigma v \rangle_-)^{-1} + (\langle \sigma v \rangle_+)^{-1} \simeq (\langle \sigma v \rangle_{thermal})^{-1}$,

---

[8]More conservatively, imposing $\Delta N_{eff} < 0.3$ allows up to five dark photons. This requires $g_{SM}^{\star s}(T^{dec}) > 29, 49, 67, 83, 98$ and correspondingly $T^{dec} \gtrsim 0.15, 0.25, 0.6, 9, 65$ GeV, for 1, 2, 3, 4, 5 dark photons, respectively.

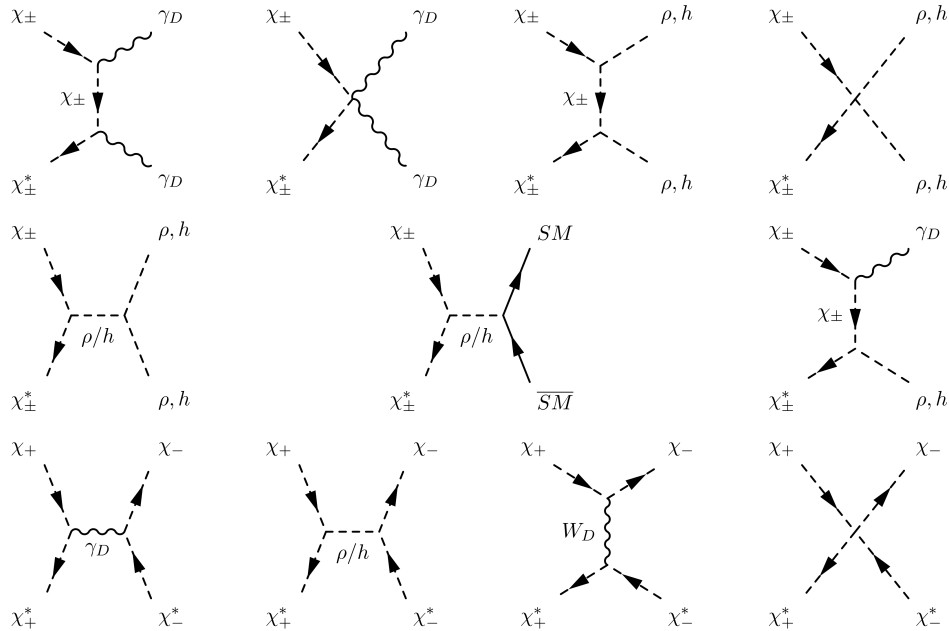

Figure 1: Annihilation processes for the DM candidates $\chi_\pm$ in the $SU(2)_D$ model.

where $\langle\sigma v\rangle_{thermal} \simeq 3 \cdot 10^{-26}$ cm$^3$/s refers to the usual thermal freeze-out cross section for one DM species. As the relic densities $\Omega$ are inversely proportional to $\langle\sigma v\rangle$, one has $\Omega_{\chi_+}/\Omega_{\chi_-} \simeq m_{\chi_+}^2/m_{\chi_-}^2$. This also fixes $\alpha_D$ as a function of $m_{\chi_\pm}$, according to $\alpha_D^2 \simeq 3.5 \cdot 10^{-9}(m_{\chi_+}^2 + m_{\chi_-}^2)/$GeV$^2$.

A more precise calculation can be obtained by taking into account the fact that $x_f \equiv m_\chi/T_f$, where $T_f$ is the freeze-out temperature, will not be exactly the same for $\chi_+$ and $\chi_-$, and for a single-component DM scenario. Taking into account this effect we obtain $x_f^-(\langle\sigma v\rangle_-)^{-1} + x_f^+(\langle\sigma v\rangle_+)^{-1} \simeq x_f(\langle\sigma v\rangle_{thermal})^{-1}$, where the three values of $x_f$ are determined as usual from the decoupling condition, see for instance Eq. (32) in [4]. Using this relation, Fig. 2 shows, as a function of $m_{\chi_-}$ and for various values of the mass ratio $m_{\chi_+}/m_{\chi_-}$, the value of $\alpha_D$ which accounts for the observed DM relic density. Note that we neglected the effect of the annihilation $\chi_+\chi_+^* \to \chi_-\chi_-^*$. We estimated that it modifies the result in Fig. 2 by at most 0.2%.

As for a generic thermal candidate, a DM mass of order $\sim$ TeV is required if the couplings driving the annihilation are of order unity. The DM mass in this hidden sector freeze-out scenario is bounded both from above and from below. On the one hand, there is a unitarity constraint on the cross section, $\langle\sigma v\rangle_J \lesssim 4\pi(2J+1)/(m_{DM}^2 v)$, where $v$ is the relative velocity [39]. As the partial-wave expansion of the cross section is dominated by the S-wave, Eq. (44), the relevant bound is obtained taking $J = 0$ and $v^2 = 3x_f^{-1}/2 \simeq (0.3)^2$ for $x_f \simeq 20$ [40]. When applied to Eq. (44), unitarity gives a bound on the coupling strength driving the annihilation, $\alpha_D \lesssim 7.3$. Together with the relic density constraint this results in an upper bound on the mass of the heaviest DM component, $m_{\chi_+} \lesssim 100$ TeV.

On the other hand, the dark photons should decouple from the SM at a temperature large enough to satisfy the extra radiation constraint $\Delta N_{eff} < 0.135$, see section 5.1. Since the decoupling occurs when the $\chi_-$ decouples, at $T^{dec} = m_{\chi_-}/x_f^-$ with $x_f^- \sim 20$, the lower bound is $m_{\chi_-} > T_{min}^{dec}x_f^-$. We thus obtain an absolute lower bound $m_{\chi_-} \gtrsim 6$ GeV, see Fig. 2 in the case $m_{\chi_+}/m_{\chi_-} = 1$. For a larger mass ratio the lower bound is slightly larger, as $x_f$ slowly grows.

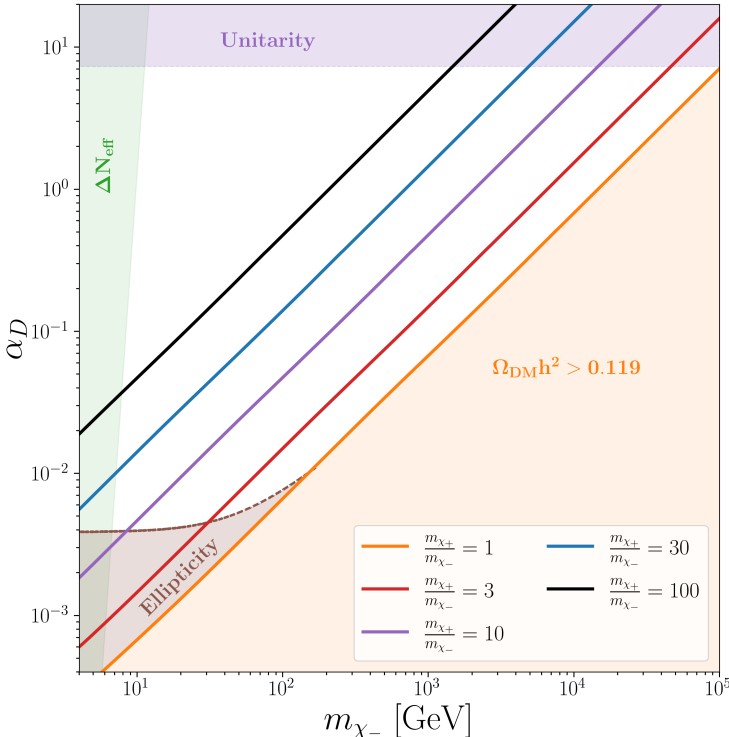

Figure 2: Values of $\alpha_D$ leading to the observed DM relic density, in the dark photon annihilation regime, as a function of $m_{\chi_-}$, for various values of the ratio $m_{\chi_+}/m_{\chi_-}$. The regions forbidden by the unitarity, extra radiation, and ellipticity constraints are also shown, see text.

Note that, in the figure, the $\Delta N_{eff}$ bound applies only assuming the observed relic density is reproduced, i.e. only along the solid line for each given mass ratio.

Note finally that the long range force driven by $\alpha_D$ may affect the formation of structures, in particular galactic scale structures. This so-called ellipticity constraint is obtained by requiring that the timescale for the DM halo to become isothermal, i.e. for an average DM particle to change its kinetic energy by a factor of $\mathcal{O}(1)$ through Coulomb interactions, is larger than the age of the Universe [41–43]. Such an isothermal DM halo is thought to lead to a spatial isotropisation of the matter distribution over a similar timescale, erasing any elliptic feature. This constraint gives an upper bound on the strength of the long range force, $\alpha_D \lesssim 1.6\sqrt{10^{-11}(m_{\chi_+}/\text{GeV})^3}$ for $m_{\chi_+} >> m_{\chi_-}$ [42,43] (for scalar DM with charge $Q_D = 1/2$ as in our scenario). For $m_{\chi_+} \sim m_{\chi_-}$, there are extra diagrams involving both components contributing to the DM scattering, but in first approximation the bound is given by this same expression. The intersection of the upper edge of the brown shaded region in Figure 2 with the various relic-density lines shows the lower bound on $m_{\chi_-}$ for the corresponding mass ratio $m_{\chi_+}/m_{\chi_-}$, under the assumption that the freeze-out process is dominated by the annihilation into dark photons, according to Eq. (44). As a result only the relatively small brown shaded region is excluded by ellipticity. For $m_{\chi_+} = m_{\chi_-}$ the bound is $m_{\chi_+} \gtrsim 180$ GeV. A relaxation of this delicate constraint by a factor of 3 for $\alpha_D$ results in a milder bound, $m_{\chi_+} \gtrsim 20$ GeV. As the ratio $m_{\chi_+}/m_{\chi_-}$ increases, the lower bound on $m_{\chi_-}$ largely decreases as shown in the figure, while the one on $m_{\chi_+}$ slowly decreases, down to $\sim 80$ GeV in the limit where $m_{\chi_+} >> m_{\chi_-}$. Combining both ellipticity and $\Delta N_{eff}$ constraints gives $m_{\chi_-} \gtrsim 7$ GeV. We refer the reader to [41–45] for more discussions on the ellipticity constraint.[9]

---

[9]One should indeed be careful with this bound. As noted by the Authors of [42], the assumption that the DM ve-

### 5.2.2  DM annihilation into dark scalars

The ellipticity bound discussed above holds only if the DM annihilation is dominated by the coupling to massless dark photons. If instead a short range interaction dominates, such as the coupling to the massive dark scalar $\rho$, the ellipticity constraint becomes irrelevant, since the value of $\alpha_D$ can be small in this case. As a result, smaller DM masses become allowed.

Let us consider, indeed, the regime along which the DM annihilation proceeds dominantly into a pair of lighter $\rho$ particles. Such annihilation can be induced by the trilinear $\kappa$ and/or the quartic $\lambda_{\chi\Phi}$ scalar interactions, see the third, fourth and fifth diagram in Fig. 1. There is also the possibility to induce $\chi$'s annihilations into $\rho$'s by a transition into SM Higgs bosons (from the $\lambda_{\chi H}$ interaction), which in turn mix into $\rho$ bosons (from the $\lambda_{\chi\Phi}$ interaction).

Here we restrict ourselves to a minimal case where the DM annihilation into a pair of $\rho$ particles is dominated solely by the $\lambda_{\chi\Phi}$ coupling. Neglecting $m_\rho$ with respect to the DM masses $m_{\chi_\pm}$, the relevant cross section reads

$$\langle \sigma_{\chi_\pm \chi_\pm^* \to \rho\rho} v \rangle = \frac{\lambda_{\chi\Phi}^2}{64\pi m_{\chi_\pm}^2} \left( \lambda_{\chi\Phi}^2 \frac{v_D^4}{m_{\chi_\pm}^4} - 2\lambda_{\chi\Phi} \frac{v_D^2}{m_{\chi_\pm}^2} + 1 \right). \tag{45}$$

We assume that the first term in brackets, corresponding to the third diagram of Fig. 1, is dominant,[10] which implies $\Omega_{\chi_+}/\Omega_{\chi_-} \simeq m_{\chi_+}^6/m_{\chi_-}^6$.

Let us define the dimensionless effective couplings $\lambda_\pm \equiv (\lambda_{\chi\Phi} v_D)/m_{\chi_\pm}$. The Fig. 3 shows the value of $\lambda_+$ needed to account for the DM relic density, as a function of $m_{\chi_-}$.[11] The general pattern is relatively similar to the one of the dark photon annihilation regime of Fig. 2, in particular larger $m_{\chi_-}$ and/or larger $m_{\chi_+}/m_{\chi_-}$ require larger $\lambda_+$. The unitarity constraint has nevertheless a different shape in the two figures, but this is just an artefact of the choice of the coupling on the $y$-axis in Fig. 3. Had we plotted $\lambda_-$ instead of $\lambda_+$, the unitarity constraint would be a horizontal line similarly to Fig. 2. Such constraint is obtained from the $\chi_-$ annihilation cross section, as it is larger than the $\chi_+$ one. In virtue of the absence of the ellipticity constraint, we observe that the value of the DM mass $m_{\chi_-}$ can be as small as $\sim 10$ GeV, see Fig. 3 for $m_{\chi_+}/m_{\chi_-} = 1$.

Note that viability of this scenario requires that, after DM freeze-out, the $\rho$ particles transfer their energy into the SM thermal bath (not to overclose the Universe). This can be simply achieved by considering a small value of the $\lambda_{\Phi H}$ coupling, leading to $\rho$ decays into SM particles through the $\rho$-$h$ mixing (with possibly little impact on the DM annihilation cross section).[12]

### 5.2.3  DM annihilation through the Higgs portal

The dominant DM annihilation channel could also be into SM particles, driven by the Higgs portal coupling $\lambda_{\chi H}$, with a negligible effect from other dark-sector interactions. The anni-

---

locity distribution (from which one infers the DM energy transfer rate) traces the matter density distribution (from which one measures the ellipticity) is somewhat fragile, and further simulations are needed to better understand the interplay between baryonic matter and self-interacting DM in shaping the halo [44,45].

[10]In fact we already assumed that $\lambda_{\chi\Phi}$ dominates over other couplings, and that the SSB scale in the dark sector, $v_D$, is sufficiently large to keep $W_D$ heavier than the DM particles $\chi_\pm$. We also take $v_D$ sufficiently large to guarantee that the contribution of the trilinear coupling $\kappa$ to DM annihilations (proportional to $\kappa^4$) will be subleading, even though $\kappa$ is needed to generate a mass splitting between $\chi_+$ and $\chi_-$, see Eq. (22).

[11]In the plot we neglect again the effect of the process $\chi_+\chi_+^* \to \chi_-\chi_-^*$ (in this case mediated by a $\rho$). We estimated that this process modifies at most by 3% the required value of $\lambda_+$.

[12]Note that the decay of $\rho$, if it happens when it is non-relativistic, can largely reheat the SM thermal bath and consequently dilute the DM and dark photon number densities, which would relax the $\Delta N_{eff}$ constraint, and require a smaller DM annihilation cross section, in order to have less Boltzmann suppression of its number density [46–48]. This would allow for both smaller and much larger values for the DM mass. We will not consider this possibility further.

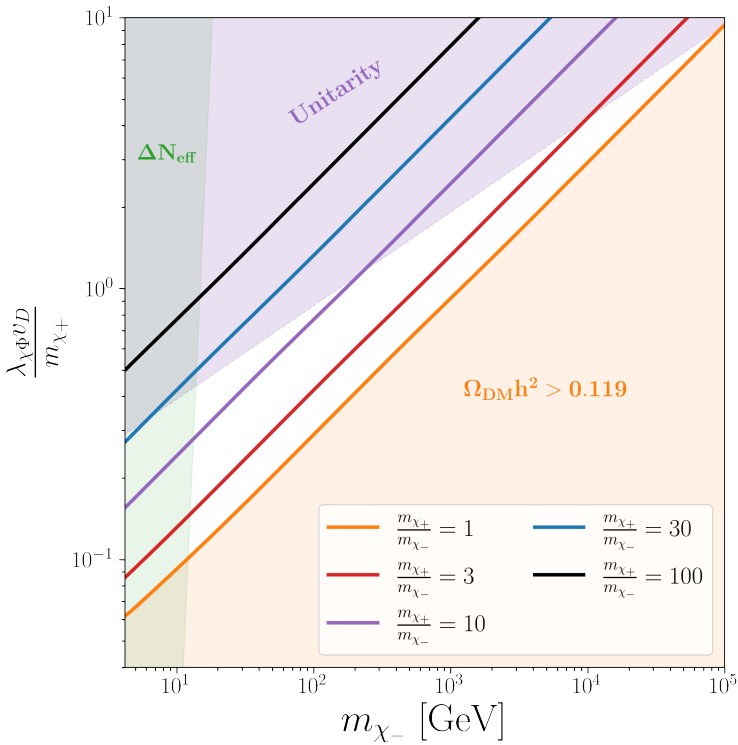

Figure 3: Values of $\lambda_{\chi\Phi}v_D/m_{\chi^+}$ leading to the observed DM relic density in the annihilation into $\rho$ regime, as a function of $m_{\chi_-}$, for various values of the ratio $m_{\chi_+}/m_{\chi_-}$. The regions forbidden by the unitarity and extra radiation constraints are also shown.

hilation final states, in this case, are a pair of Higgs bosons, $hh$, or a pair of SM particles through $h$ exchange, see the third to sixth diagrams in Fig. 1 (dropping all $\rho$ particles in these diagrams). This regime is comparable to ordinary Higgs portal DM setups (see for instance Fig. 1 of Ref. [49] for the relic density constraint on various Higgs portal couplings), except that, in our $SU(2)_D$ model, there are two complex DM scalars $\chi_\pm$, both contributing to the relic density. Specifically, both states communicate to the SM through the same Higgs portal interaction $\lambda_{\chi H}$.

The analytical form of the cross sections for the various SM final states can be found e.g. in Eqs. (B.13)-(B.16) of [50].[13] In the high energy regime, $m_{\chi_\pm} \gg m_h$, the cross sections scale as $1/m_{\chi_\pm}^2$. Therefore, the DM relic density is dominated by the $\chi_+$ component, similarly to what happens in the $\gamma_D$ and $\rho$ annihilation scenarios discussed above, with $\Omega_{\chi_+}/\Omega_{\chi_-} \simeq m_{\chi_+}^2/m_{\chi_-}^2$. In contrast, in the low energy regime, $m_{\chi_\pm} \ll m_h$, the dominant cross section, is into two SM fermions $f$ and scales as $m_f^2/m_h^4$. As a result, the lightest $\chi$ component will tend to dominate the relic density because, it tends to annihilate to lighter fermions and thus has a smaller annihilation cross section at freeze-out. The left panel of Fig. 4 shows, for an example value of the mass ratio $m_{\chi_+}/m_{\chi_-} = 10$, how the transition from dominant $\Omega_{\chi_-}$ to dominant $\Omega_{\chi_+}$ occurs.

In the right panel of Fig. 4 we show the values of $\lambda_{\chi H}$ one needs to account for the relic density. For the sake of comparison, we also show the curve for the case where DM is an SM-singlet real scalar $s$, with portal $\mathcal{L}_{portal} = -(\lambda_{sH}/2)s^2 H^\dagger H$. For fixed, large value of $m_{\chi_-}$, due to the scaling of the annihilation cross sections $\propto 1/m_{\chi_\pm}^2$, as the mass of the dominant state $m_{\chi_+}$ increases, a larger Higgs portal is required. Remarkably, this means that this scenario

---

[13]These equations hold for a real scalar DM candidate. For a complex scalar DM particle such as $\chi_+$ or $\chi_-$, the combinatorial factors differ, which implies that the cross sections are obtained from the ones of [50] by replacing the $\lambda$ Higgs portal coupling of [50] by $\lambda_{\chi H}/2$.

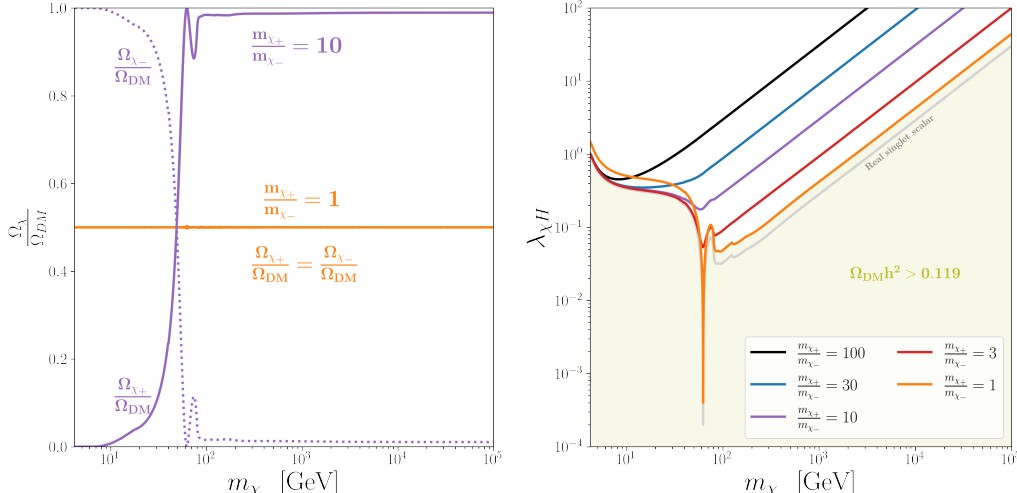

Figure 4: The DM relic density in the Higgs portal regime. The left panel shows the values of $\Omega_{\chi_\pm}/\Omega_{DM}$, for $m_{\chi_+}/m_{\chi_-} = 10$ (purple) and for $m_{\chi_+}/m_{\chi_-} = 1$ (orange). The right panel shows the values of the Higgs portal coupling $\lambda_{\chi H}$ leading to the observed DM relic density, as a function of $m_{\chi_-}$, for various values of the ratio $m_{\chi_+}/m_{\chi_-}$. We shaded the region where the DM relic density is too large, independently of the mass ratio. For the sake of comparison we also show the coupling needed when DM is an SM-singlet real scalar.

predicts a light state $\chi_-$ with a Higgs portal interaction larger than in the minimal real-scalar-singlet DM model. This is possible because such light state has a subdominant relic density. In the opposite limit of small $m_{\chi_-}$, as soon as $m_{\chi_+}/m_{\chi_-} \gg 1$, the exact value of this ratio becomes irrelevant, since the contribution of $\chi_+$ to the relic density is negligible. Note that, for large values of the mass ratio, there is no more resonant enhancement of the relic density when the DM mass approaches $m_h/2$, as in the ordinary DM Higgs portal scenario. This is because, when $m_{\chi_+} \simeq m_h/2$ it is $\chi_-$ that dominates the relic density, and vice versa.[14]

In Fig. 5 we show the various constraints which hold on the Higgs portal interaction, for three representative values of the mass ratio, $m_{\chi_+}/m_{\chi_-} = 1, 10, 100$. Below the Higgs resonance, there is a stringent constraint from the Higgs invisible decay width, i.e. $\mathrm{Br}_h^{inv} < 0.13$ at 95% CL [51]. As can be seen in the case $m_{\chi_+}/m_{\chi_-} = 10$, the invisible-width constraint has a jump when the channel $h \to \chi_+ \chi_+^*$ becomes kinematically forbidden (and in all cases it disappears when $h \to \chi_- \chi_-^*$ is kinematically forbidden too). This constraint rules out the possibility that DM annihilation through the Higgs portal would be dominant below the Higgs resonance.

As already explained, above the resonance and for $m_{\chi_+}/m_{\chi_-}$ significantly larger than unity, the Higgs portal regime predicts a subleading DM component, $\chi_-$, lighter than the dominant component, $\chi_+$, with a large Higgs portal interaction. Remarkably, in this case the direct-detection constraint also relaxes, with respect to a single DM scalar scenario with mass equal to $m_{\chi_-}$, because the $\chi_-$ DM flux in the detector is suppressed. This is illustrated in Fig. 5, where

---

[14]In both panels of Fig. 4 we took into account the process converting $\chi_+$ pairs into $\chi_-$ pairs, which here is dominated by Higgs-boson exchange (ninth diagram in Fig. 1). This process can largely suppress the $\chi_+$ relic density when $m_{\chi_+} \lesssim m_W$, because in this case the $\chi_+$ conversion into $\chi_-$ is more efficient than the annihilation into SM pairs, which is suppressed by small Yukawa couplings. However, this does not affect the value of the Higgs portal coupling shown in Fig. 4 by more than $\sim 40\%$, because this process leaves the $\chi_-$ relic density unchanged, in good approximation. When $m_{\chi_+}$ and $m_{\chi_-}$ are different but very close (not shown in the figure), the effect could be more important. We do not explore further this possibility, because the low-mass region is anyway excluded by collider and direct detection constraints, as described below.

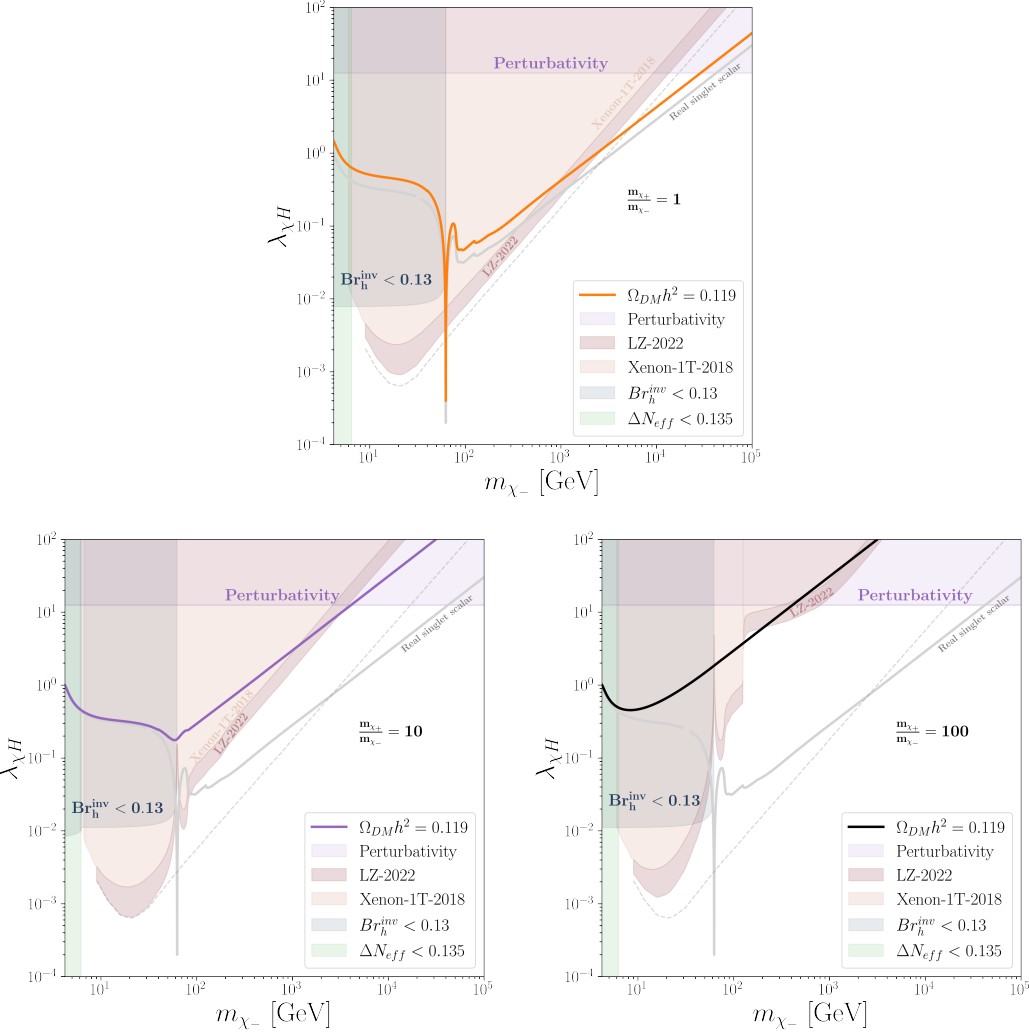

Figure 5: Constraints holding on the DM annihilation through the Higgs portal, for three values of the mass ratio $m_{\chi_+}/m_{\chi_-} = 1$ (upper panel), 10 (lower left), and 100 (lower right): Higgs-boson invisible width [51] (excluding the dark blue shaded region), DM direct detection [52, 53] (light and dark brown, for Xenon and LZ respectively), perturbativity (purple), and $\Delta N_{eff}$ (green), see text. In each panel, the solid curve (already shown in Fig. 4) indicates the values of $\lambda_{\chi H}$ which reproduce the observed DM relic density. For the sake of comparison we also show the coupling needed when DM is an SM-singlet real scalar (solid grey line) and the corresponding direct detection constraint [53] (dashed grey line).

we show the Xenon1T [52] as well as the very recent LZ [53] constraints: taking into account the value of $\Omega_{\chi_+}/\Omega_{\chi_-}$ predicted by the annihilation cross sections, one observes that these constraints, for DM masses above the Higgs resonance, become rapidly weaker for growing $m_{\chi_+}/m_{\chi_-}$. Note that, above the resonance, the direct detection bound comes from the non-detection of $\chi_-$ particles. Even if the $\chi_-$ component is subleading, the constraint on the direct detection of the $\chi_-$ flux is indeed stronger than on the direct detection of the larger $\chi_+$ flux. This is because the DM elastic cross section on nuclei scales as $1/m_{\chi_\pm}^2$ (see e.g. [54]), and because the direct detection upper limit on the elastic cross section relaxes when the mass increases. Note also that when the annihilation channel $\chi_- \chi_-^* \to h h$ opens, the cross section of the subleading $\chi_-$ component suddenly increases, and thus the $\chi_-$ relic density suddenly

decreases, which explains why the direct detection constraints become suddenly weaker.[15]

Combining these effects, we find that, as the mass ratio varies, the sensitivity of direct detection searches remains close to the coupling values needed for the correct relic density, see the three panels of Fig. 5 in the region $m_{\chi_-} \sim$ TeV. In the case $m_{\chi_+}/m_{\chi_-} = 1$ (upper panel), direct detection implies a DM mass above the TeV scale, $m_{\chi_-} \gtrsim 2$ TeV, while the perturbativity requirement, $\lambda_{\chi H} \lesssim 4\pi$, implies an upper bound $m_{\chi_-} \lesssim 30$ TeV. The case $m_{\chi_+}/m_{\chi_-} = 10$ (lower left panel) is very close to be excluded. In the case $m_{\chi_+}/m_{\chi_-} = 100$ (lower right panel) there is a small allowed region at $m_{\chi_-} \simeq 200$ GeV. All in all, as direct detection experiments are expected to make progress in a near future, they could soon exclude the Higgs portal regime of our $SU(2)_D$ model, or observe a signal at the level of the present sensitivity. All the regions which are still allowed in Fig. 5 would be fully covered by any new direct detection experiment which could improve by a factor $\sim 10$ the sensitivity on the DM-nucleon elastic-scattering cross section (with respect to the LZ constraint showed in this figure). This improvement is within reach of several proposed or under-construction experiments [55–57]. Furthermore, the observation of a positive signal in the $m_{\chi_-} \simeq 200$ GeV region just discussed above would, given the Higgs portal coupling of order one it involves, open the possibility to test this kind of scenario at future colliders in the long term (see for instance [58, 59]).

### 5.2.4 Non-thermal DM regimes

The discussion above assumes that the Higgs portal interactions are not tiny, so that the dark sector and the SM thermalise at high temperatures prior to freeze-out. If this is not the case, other DM production regimes can account for the relic density.

The most straigthforward is a freeze-in dominated by the $\lambda_{\chi H}$ Higgs portal coupling, inducing a direct $\chi$-pair production from the annihilation of two SM particles. This requires values of the portal coupling of order $10^{-11}$, see for instance Fig. 15 of Ref. [4]. Above the $Z$ threshold, for $m_{\chi_-} > m_Z/2$, this leads to $\Omega_{\chi_+} \simeq \Omega_{\chi_-}$, as a result of the compensation of two effects: on the one hand, for $m_{\chi_-} < m_{\chi_+}$ one $\chi_-$ contributes to $\Omega_{DM}$ less than one $\chi_+$; on the other hand, more $\chi_-$ are created than $\chi_+$, since the freeze-in production, which is infrared dominated, stops at about the mass of the DM particle created.

Another possibility is freeze-in dominated by the $\lambda_{\Phi H}$ Higgs portal interaction, along which a pair of SM particles can produce a pair of $\chi_\pm$, $W_D$ or $\rho$ particles. Once produced, the dark sector particles decay to the lightest available final state. As usual, for $m_{\chi_+} + m_{\chi_-} < m_{W_D}$, one is left with a DM population made by $\chi_\pm$ particles. It is beyond the scope of this paper to analyse quantitatively this indirect freeze-in possibility, which typically requires a $\lambda_{\Phi H}$ of order $10^{-11}$ as well.

For values of the Higgs portal couplings larger than for freeze-in, one might create enough dark particles from the SM to have thermalisation within the dark sector, even though the portals are still small enough to prevent thermalisation of the two sectors. In this case the relic density can also be produced, in the secluded freeze-out or in the reannihilation regime, see Fig. 13 of [4]. Reannihilation of $\chi_\pm$ occurs when $m_{\chi_\pm}$ is larger than few GeV: the out-of-equilibrium DM production from a pair of SM fermions (driven by heavy quark Yukawa or gauge couplings) is still active [4] when the annihilation $\chi\chi \to \gamma_D\gamma_D$ goes out-of-equilibrium, at a dark-sector temperature $T_D^f \simeq m_{\chi_\pm}/20$. For lower $\chi_\pm$ masses the secluded freeze-out regime will in general hold: in this case the Higgs portal DM pair production (suppressed by small SM Yukawa couplings) stops being active before the annihilations freeze within the dark-sector thermal bath.

---

[15]In a similar vein note also that close to the resonance $m_{\chi_-} \simeq m_h/2$, the direct detection constraint features a peak behaviour, because the still dominating $\chi_-$ component rapidly decreases due to the resonance.

### 5.3 $SU(3)_D$ relic density and constraints

#### 5.3.1 General case

In the $SU(3)_D$ model the discussion of the relic density is relatively similar to the one for the $SU(2)_D$ model, provided the $\chi$ is the lightest stable particle in the dark sector, apart from the dark photons. In particular, the three thermal regimes considered in sections 5.2.1-5.2.3 for the $SU(2)_D$ model are also reproduced, qualitatively, in the $SU(3)_D$ model.

There are nevertheless important differences. In the $SU(3)_D$ case there is no mass splitting between the three components of the DM multiplet $\chi$. Thus, there is only one annihilation cross section for the whole DM triplet. Moreover, there are regions of parameters with additional DM candidates besides $\chi$: the massive gauge boson triplet $W_D$, which can become stable if light enough (in analogy with the $SU(2)_D$ model), as well as other candidates, specific to the $SU(3)_D$ model, i.e. the scalar triplets $\tau$ and $\varphi$ contained in the ten-plet $\Phi$. They could also be stable if light enough, leading to alternative DM scenarios, that we do not examine in details here.

Even sticking to the minimal case of $\chi$ triplet DM, the $SU(3)_D$ model allows for additional processes that can play a role in the DM production, in particular the semi-annihilation process associated to the $Z_3$ triality symmetry, that we discuss next.

#### 5.3.2 Semi-annihilations from triality and consequences for DM detection

The triality symmetry $Z_3$ of the $SU(3)_D$ model has a clear phenomenological signature: DM particles can undergo semi-annihilations [5,60,61], $\chi\chi \to \chi^* X$. The crucial interaction is $\kappa$ in Eq. (16), which couples three DM $\chi$ triplets to one scalar ten-plet $\Phi$. Taking also into account that $\Phi$ acquires a VEV, there are several possibilities for the final-state particle $X$: it can be either one of the $\Phi$ components, or a dark gauge boson $W_D$ or $\gamma_D$ (by one insertion of the dark gauge coupling $g_D$), or a Higgs boson (by one insertion of the portal coupling $\lambda_{\Phi H}$ or $\lambda_{\chi H}$).

If the mass of the $X$ particle is sizeably smaller than $m_\chi$, the semi-annihilations lead to a flux of semi-relativistic DM particles from the Galactic centre, or from the centre of the Sun or the Earth. As these DM particles are boosted, there is a possibility of DM direct detection in neutrino telescopes [62], via elastic scattering of the monochromatic flux of DM particles on nucleons or electrons. So far, to our knowledge, only the Super-Kamiokande experiment reported a search for such particles [63], setting upper limits on the coupling strength of a specific model. It would be worth repeating the same analysis at Super-Kamiokande and higher-energy neutrino telescopes, for the case of a Higgs portal mediated elastic scattering, corresponding to our $SU(3)_D$ model.

It is interesting to note that, in the $SU(3)_D$ model, there is a regime where the semi-annihilation process can largely dominate the freeze-out (this was not the case, for example, in the original semi-annihilation model [5], but can happen in other semi-annihilating frameworks [64]). This can occur in particular if all couplings are small with respect to the semi-annihilation coupling $\kappa$ in Eq. (16). Such situation is interesting because it would not only maximise the flux of boosted DM particles for direct detection in neutrino telescopes, but it also leads to a definite prediction for the value of this flux. Note that, since ordinary DM annihilations $\chi\chi^* \to SM\,SM$ are suppressed in this regime, the searches for DM scattering on nuclei are correspondingly harder.

To discuss this possibility, let us consider the simple case where DM semi-annihilates only into a Higgs boson. This applies when DM consists exclusively of the $\chi$ triplet, with all other massive dark-sector particles heavier, and when all couplings are small except for $\kappa$ and $\lambda_{\Phi H}$. In particular for $g_D$ sufficiently small the semi-annihilations into $\gamma_D$ during the freeze-out are

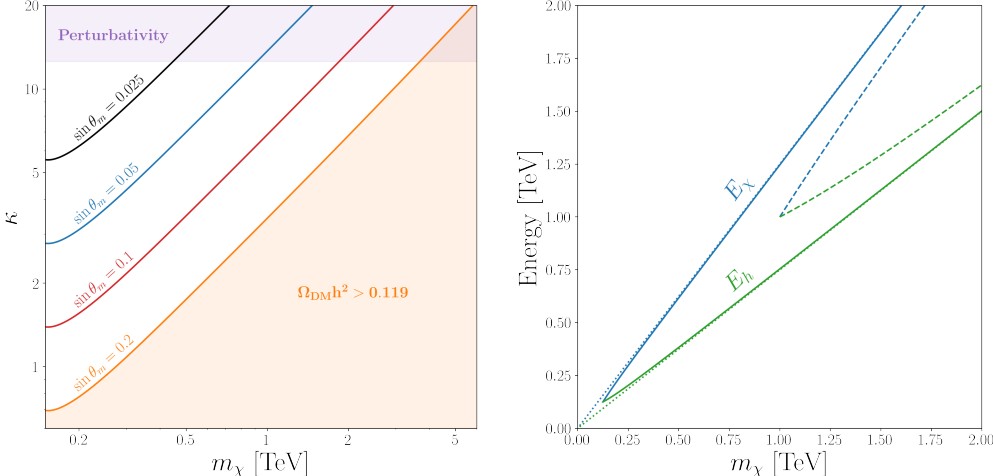

Figure 6: Left panel: Values of the trilinear coupling $\kappa$ leading to the observed DM relic density, in the regime where the semi-annihilations $\chi\chi \to \chi^* h$ are dominant, as a function of $m_\chi$ and for different values of the $h - \rho$ mixing angle $\theta_m$. Right panel: The solid (dotted, dashed) curves show the energy of the $\chi$ (blue) and $X$ (green) monochromatic lines as a function of $m_\chi$, for $X = h$ with $m_h = 125$ GeV ($X = A_D$ with $m_{A_D} = 0$, $X = \rho$ with $m_\rho = 1$ TeV).

suppressed.[16] In this case the flux of boosted DM consists of a monochromatic flux of DM particles with energy $E_\chi = (5m_\chi^2 - m_h^2)/(4m_\chi)$. The energy of this 'DM line' is therefore fixed by the values of the DM and Higgs boson masses. Thus, the flux of boosted DM particles crossing the Earth is equal, for instance, to the flux of monochromatic neutrinos one obtains in scenarios where DM annihilates into a pair of neutrinos (see e.g. [65–67]).[17] The observation of such predicted flux intensity would definitely point toward a semi-annihilation origin. Moreover, in this scenario the DM and Higgs fluxes are equal, so that a characteristic correlated flux of cosmic rays from the Higgs particles is expected. The energy of the Higgs bosons produced is monochromatic and predicted to be $E_h = (3m_\chi^2 + m_h^2)/(4m_\chi)$. Its observation would allow to determine that the semi-annihilations proceed into a Higgs boson, allowing for an additional determination of the DM mass.

In Fig. 6 we show the value of the trilinear coupling $\kappa$ which leads to the observed relic density, in the regime described above where the semi-annihilations proceed into a Higgs boson and dominate the freeze-out process, for different values of the $h - \rho$ mixing angle, $\sin\theta_m \simeq \lambda_{\Phi H}(v_D v/m_\rho^2)$, induced by the Higgs portal interaction $\lambda_{\Phi H}$.[18] Indeed, in this regime the semi-annihilations into a Higgs boson are determined by the effective interaction $\mathcal{L}_{eff} \supset \sqrt{3}\,\kappa \sin\theta_m(\chi^1\chi^2\chi^3 + \chi_1^*\chi_2^*\chi_3^*)h$. The corresponding semi-annihilation cross section is

$$\langle\sigma_{ij}v\rangle \equiv \langle\sigma_{\chi_i\chi_j\to\chi_k^* h}v\rangle = \frac{9\,\kappa^2\sin^2\theta_m}{128\,\pi\,m_\chi^2}\sqrt{1 - \frac{10}{9}\frac{m_h^2}{m_\chi^2} + \frac{1}{9}\frac{m_h^4}{m_\chi^4}},\qquad(46)$$

where we retained only the s-wave contribution, which dominates for $m_\chi$ well above $m_h$. Summing over the six possible semi-annihilation channels of DM (with particles or antiparticles

---

[16]In addition, the semi-annihilation into a dark photon is not expected to lead to any sizeable boosted DM flux today, because its s-wave amplitude vanishes from conservation of spin.

[17]For example for Dirac DM annihilating dominantly into $\nu\bar\nu$ the (s-wave) annihilation produces two neutrinos, whereas a semi-annihilation produces only one DM particle, but this factor 2 is (approximately) compensated by a combinatorial factor arising from the requirement to reproduce the DM relic density.

[18]See the discussion at the beginning of section 4.1, which is qualitatively valid for the $SU(3)_D$ model as well.

in the initial state) the Boltzmann equation determining the DM number density is

$$\frac{zH(z)}{s}\frac{dY_{DM}}{dz} = -\frac{1}{6}\langle\sigma_{ij}v\rangle\left(Y_{DM}^2 - Y_{DM}Y_{DM}^{eq}\right), \tag{47}$$

where $Y_{DM} \equiv n_{DM}/s = \sum_{i=1,2,3}(n_{\chi_i} + n_{\tilde{\chi}_i})/s$, with $s$ the entropy density and $z \equiv m_\chi/T$. Note the factor of $1/6$ arising from the fact that there are 6 scalar DM states and 6 semi-annihilation channels. The left panel of Fig. 6 shows the values of $\kappa$ and $\sin\theta_m$ which correspondingly account for the observed relic density. We also display the naive perturbativity constraint $\kappa < 4\pi$, while current Higgs measurements require $\sin\theta_m < 0.2$. One observes that these constraints imply $m_\chi \lesssim 3.5$ TeV. Search for a boosted DM flux at neutrino telescopes should exclude part of the allowed parameter space, although it is difficult to make a quantitative expectation for such a search at the moment.

The right panel of Fig. 6 shows the energies of the monochromatic DM-line and Higgs-line from the semi-annihilation, according to the formulas above. Note that the model allows for multiple DM lines if the semi-annihilation does not occur only into a Higgs boson, but also into other dark-sector particles $X_i$ with energies $E_\chi^i = (5m_\chi^2 - m_{X_i}^2)/(4m_\chi)$. A couple of examples are also shown in the figure for illustration. The relative intensities of these lines depend on the mass spectrum and strength of the associated interactions. At least in principle, these multiple signatures may provide a great deal of information on the dark sector.

Besides semi-annihilations, the $SU(3)_D$ model also predicts other processes involving an odd number of DM particles, such as 2-to-3 or 3-to-2 processes: for example, a $\chi^4\chi^*$ vertex with strength $\sim \kappa\lambda_{\chi\Phi}v_D/m_\rho^2$ is induced by integrating out the $\Phi$ component $\rho$. These annihilation channels could be relevant for the relic density in specific regions of parameters, see for instance [68,69]. In a somewhat different vein, the inverse semi-annihilations $\chi^*X \to \chi\chi$ could lead to exponential production of DM particles [70].

Finally, note that both the $SU(3)_D$ and $SU(2)_D$ models are suitable to possibly induce a large, non-perturbative DM self-interaction rate, $\chi\chi \to \chi\chi$, as a result of Sommerfeld enhancement by multi-exchange of massless dark photons and/or light $\rho$ bosons. This could be relevant for the small (galactic) scale anomalies, in particular for the core-vs-cusp, too-big-to-fail and diversity problems, see e.g. [71–75]. The possibility that Sommerfeld enhancement would result from the exchange of massless mediators, rather than from massive light mediators, is a matter of debate, see e.g. [42].

## 6 Summary

We considered the simple, yet novel possibility that DM stability relies on the centre of a dark $SU(N)$ gauge symmetry. When the centre $Z_N$ is not broken spontaneously, the lightest state carrying a $Z_N$ charge is automatically stable and a DM candidate. We studied the two simplest possibilities, where the DM is a scalar $\chi$ in the fundamental representation of a $SU(2)_D$ or $SU(3)_D$ gauge group, and the gauge symmetry is spontaneously broken by a scalar $\Phi$ in the $N$-index symmetric representation (a real triplet or a complex ten-plet, respectively).

On the theory side these models are interesting in many respects. Firstly, they provide a solid ground for the existence and the stability of DM particles, which rely only on a dark gauge invariance. Secondly, the associated scalar potentials have several intriguing properties. In the $SU(2)_D$ model, beside the duality $Z_2$, there is a residual $U(1)_D$ gauge symmetry, as well as an accidental $U(1)_\chi$ global symmetry: as a consequence, not one but several particles are stable; in particular the DM doublet is split into two stable states $\chi_\pm$ which have (almost) the same couplings, but different masses. In the $SU(3)_D$ model, beside the triality $Z_3$, one is left with an unbroken $U(1)_3 \times U(1)_8$ gauge symmetry, plus a global, non-abelian discrete

group. The persistence of a non-abelian symmetry after SSB implies, in particular, that the DM behaves as a triplet, while the dark photons behave like a doublet, i.e. one predicts a massless gauge boson with four components. Note that the $N$-ality mechanism to stabilise DM is robust, and not specific to our minimal realisations: the DM stability persists in the presence of additional dark gauge symmetries, or additional fermions or scalars in the hidden sector. The only requirement is that scalar multiplets transforming non trivially under the center have a vanishing VEV.

Independently of the DM stability motivation, the scalar potential for the scalar $\Phi$ in the three-index symmetric $SU(3)$ representation has a remarkable feature. At tree-level, and in the limit where the $SU(3)$ symmetry is promoted to $U(3)$, the vacuum manifold (the set of field configurations that minimise the potential) is larger than the NGB manifold. This means there are accidental flat directions, connecting physically inequivalent field configurations: the true physical minimum and the mass for the flat directions are determined only by departures from the tree-level, $U(3)$-symmetric limit. This feature could have other interesting applications.

On the phenomenological side these models have a rich variety of specific implications. The $SU(2)_D$ model predicts the existence of two stable scalars $\chi_{\pm}$ with the same interactions but different mass. This implies a non-trivial interplay of both states, which significantly affects both the DM relic density, as well as the experimental constraints on the model. The observed relic density can be achieved along different regimes: we considered three simple and representative ones, where the dominant annihilation is into dark photons, dark light scalar bosons, and SM particles, respectively. The latter annihilation takes place through a Higgs portal interaction. In all regimes, in the viable region of parameters, the heaviest state $\chi_{+}$ always dominates the relic density, but not necessarily the various other observables which constrain the model. This implies the existence of a lighter subleading DM component $\chi_{-}$, with interactions which can be larger than would be allowed if $\chi_{-}$ were dominating the relic density.

In the dark photon annihilation regime the light (heavy) DM state must have a mass above $\sim 7$ GeV ($\sim 80$ GeV) as a result of $\Delta N_{eff}$ and ellipticity constraints. For the annihilation into dark scalars, it must lie above the $\sim 10$ GeV scale, as a result of the $\Delta N_{eff}$ constraint. For the Higgs portal regime, the direct-detection constraints allow for a large-mass window, with $m_{\chi_{-}} \gtrsim$ TeV, as long as $m_{\chi_{+}}/m_{\chi_{-}} < 10$ or so, as well as for an intermediate-mass window above the Higgs threshold, $m_{\chi_{-}} \gtrsim m_h$, which opens for larger mass ratios. The non-observation of a positive signal at near future direct-detection experiments would basically exclude the Higgs portal regime. A positive signal instead would imply a large Higgs portal interaction which could be possibly tested at future colliders.

In the $SU(3)_D$ model, the residual unbroken symmetries imply that the three components of the DM multiplet $\chi$ are degenerate in mass with the same interactions. Thus, there is no interplay of several DM components as in the $SU(2)_D$ model. On the other hand, the $SU(3)_D$ model features a 'non-abelian' pair of dark photons, which might have distinctive signatures in $N_{eff}$ and possibly elsewhere. In addition, the model predicts processes with an odd number of DM particles. This is associated to the remnant $Z_3$ triality symmetry, which allows vertices with three DM particles. Remarkably, if the DM cubic interaction is large enough, the associated semi-annihilation process(es) can dominate the freeze-out, thus fixing the value of the semi-annihilation cross section. As a consequence, one can precisely predict the position and intensity of the monochromatic flux of boosted DM, i.e. a 'DM-line', from the Galactic centre (or the Sun or Earth centre), which could be possibly observed in neutrino telescopes.

## Acknowledgements

We thank M. Ardu, G. Facchinetti, M. Hufnagel, G. Moultaka, and especially R. Argurio and F. Brümmer for useful discussions. MF thanks the SPT (ULB) for hospitality along the development of this work.

**Funding information** MF has received support from the European Union Horizon 2020 research and innovation programme under the Marie Skłodowska-Curie grant agreement No 860881-HIDDeN. NGY and TH work is supported by the Excellence of Science (EoS) project No. 30820817 - be.h "The H boson gateway to physics beyond the Standard Model", by the "Probing dark matter with neutrinos" ULB-ARC convention and by the IISN convention 4.4503.15. The work of NGY is further supported by the Communauté française de Belgique through a FRIA PhD grant.

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
