# Peer review of "Dark matter from the centre of SU (N )"

_SciPost Physics, doi:SciPost Phys. 15, 177 (2023)_

## Round 1 · Referee Report · Anonymous (Referee 1) · 2023-3-13

Strengths

  1. The paper focuses on the important question of stability of DM.
  2. The analysis is detailed and systematic.
  3. The paper is clearly written.

Report

The stability of DM is an important question that the paper focuses on. It builds a new class of models based on $SU(N)_D$ gauge groups. After symmetry breaking, a residual $Z_N$ symmetry stabilizes the DM particles. The paper is worthy of publication. However, there are a few places where the discussion could be improved.

Requested changes

  1. In sec 5.1 it would be useful to clarify that the dark sector and the SM are assumed to be in thermal equilibrium at very early times, if that is the assumption the authors are making to compute $\Delta N_{\rm eff}$.
  2. On page 17, just above sec 5.2.2., it would be useful if the authors expand on the `ellipticity constraint' for the discussion to be self contained.
  3. In sec 5.3.2, the authors describe various signatures from semi-annihilations. It would be useful to know whether some of those constraints actually constrain the parameter space in Fig. 6 where only overclosure and unitarity bounds are shown.
  4. In Fig. 5 top panel, there is a region of parameter space where relic density can be successfully explained without running into other constraints. I am wondering whether the authors could briefly comment on whether and to what extent future direct detection experiments can be sensitive to the remaining parameter space on the orange line.

---

## Round 1 · Referee Report · Anonymous (Referee 2) · 2023-3-20

Report

The origin of DM stability is a key issue in DM model buildings and phenomenology. This manuscript addresses this issue, assuming that DM is stable due to N-ality of non-Abelian dark SU(N) gauge symmetry. This is a generalization of Krauss-Wilczek mechanism of $U(1)$ gauge symmetry to the non-Abelian cases and makes an interesting possibility for DM stability. The materials in the manuscript are original, useful and interesting enough. The manuscript is well organized with no grammatical problem. Most parts of the manuscript are clearly written. The manuscript is worth to be considered for publication in SciPost. Before I recommend the acceptance of this paper, I would suggest the authors address some minor issues that are described below.

  1. In the overview of the earlier works in the Introduction, the authors review earlier literature on the DM stability in terms of gauge symmetry. However, there is no clear distinction whether DM is stabilized by global or gauge symmetry (I guess they meant local gauge symmetry). I would suggest that the authors make two categories of DM models more clearly, where DM is stable (or long lived) because of global or local dark gauge symmetry and add relevant references according to these two categories. For example, Ref. [9] is about scalar DM models, based on global $Z_2$ and $Z_3$ symmetries. Local symmetry versions of $Z_2$ and $Z_3$ (both model construction and detailed phenomenology) were presented in detail, for example, in https://arxiv.org/abs/1407.6588 and https://arxiv.org/abs/1402.6449, respectively. Underlying philosophy and DM phenomenology in DM models with global and local dark gauge symmetries are vastly different in terms of particle contents: dark Higgs boson and dark photon. And these two categories of global vs. local dark gauge symmetries are better to be clearly distinguished. There could be similar cases in Introduction and other sections, and appropriate changed are recommended.

  2. The main motivation for this manuscript is the origin of DM stability. But DM decays from dim-5 and dim-6 operators induced by gravity are not addressed clearly in case of dark symmetry is global. For example, in https://arxiv.org/abs/1303.4280, it was noted that global symmetry would be generically violated in the presence of gravity, and electroweak mass scale DM would decay too fast through gravity-induced dim-5 operators, and cannot make a good DM of the Universe. This problem could be evaded if one considers local dark gauge symmetry, instead of global dark symmetry. It would be nice to include this discussion in Introduction, since it will strengthen the importance of the proposal made in this manuscript.

  3. In Sec. 3.1, the authors discuss in brief monopole DM, summarizing the work by Murayama and Shu, who discussed topological monopole DM without explicit construction of the model. The explicit model was constructed in Ref. [19] ( https://arxiv.org/abs/1311.1035 ), and it was shown that multi-component dark sector with massive vector DM, monopole DM and massless DR arises from spontaneous dark gauge symmetry breaking, $SO(3)_D \rightarrow U(1)_D$. In the manuscript, Ref.[19] is cited only in the context of massive vector DM. It would be nice to mention [19] in the paragraph on monopole DM in the earlier part of Sec. 3.1.

  4. Sec. 4.2 and Sec. 5.3 discuss the case $SU(3)_D$ broken to $U(1)_3 \times U(1)_8$. In this case, there is no self-interaction among the DR, and no constraints from large scale structure (see, for example, [1609.02307] Residual Non-Abelian Dark Matter and Dark Radiation (arxiv.org) , which could be cited along with Refs. [18,19,20,21] since the model there is qualitatively different from Refs.[18-21]). If the unbroken gauge group is non-Abelian as described in the above reference, the gauge coupling should be very tiny and thermal WIMP scenario or SIDM cannot be realized. Otherwise, there would be too much suppression of the matter power spectrum in small scales (large $k$ region), in disagreement with the observation. It would be important and interesting to know if the symmetry breaking patterns for $SU(N>3)$ can be non-Abelian or simply products of U(1)’s. If the latter is the case, the proposal in this manuscript would get another strong point, since one can avoid stringent constraints from cosmology on matter power spectrum.

---

## Round 2 · Referee Report · Anonymous · 2023-4-24

Report

Most of the comments and suggestions were properly addressed, and I am happy to recommend the acceptance of the manuscript for publication in SciPost.

---

## Round 2 · Author Response

Here is our answer to the Referees and the corresponding changes to the manuscript:

Report 1:

We thank the Referee for useful comments, and we addressed the various points raised as follows:

  1. We added the required sentence at the very beginning of section 5.

  2. Concerning the ellipticity constraint, in the last paragraph of section 5.2.1 we added a couple of sentences to explain better how this constraint is obtained. We provide appropriate references for the Readers interested into a more technical discussion.

  3. We agree with the Referee that it would be useful to derive such constraints, but we are not aware of any dedicated study at the moment, which could allow to specify quantitatively which part of the left panel of Figure 6 could be probed by neutrino telescope searches for a boosted DM flux. A few lines below eq. 47 we added a sentence to say this explicitly.

  4. At the end of section 5.2.3 we added a few sentences to state explicitly that the next generation of direct detection experiments will be able to exclude the allowed regions in figure 5.

Report 2:

We thank the Referee for useful comments and pointing out relevant references in his report. Here are our answers to the specific points raised:

1&2. In this paper, we decided to specifically address DM that is stabilised by gauge symmetries only. In the introduction we review the literature specific to this type of mechanism, so we do not discuss stabilisation through global symmetries. We amended our references according to the suggestions of the Referee.

  1. We added the citation to the paragraph discussing monopole DM as requested.

  2. Models with SU(N) gauge symmetry with N>3 are problematic, as we already stated in section 2. In addition, to determine the symmetry breaking pattern would require an involved study of the properties of the potential for each case. The large scale structure constraints mentioned by the Referee are tricky and do not apply to our models with N=2,3, therefore we prefer not to enter into this discussion in this paper.

---

## Round 2 · List of Changes

See our answer to the Referees.

---

## Editorial Decision

published